# Cell-specific mechanisms drive connectivity across the time course of Huntington's disease

Carlos Estevez-Fraga [1], Isaac Sebenius [2,3], Justine Y. Hansen [4], Benjamin Hänisch[5,6,7], Paul Zeun[8], Rachael I. Scahill[1], Sarah Gregory[1], Eilanoir B. Johnson[9], Edward J. Wild[1], Lauren M. Byrne [1], Alexandra Durr [10], Bernhard Landwehrmeyer[11], Blair R. Leavitt [12,13], Bratislav Misic [4], Sofie Louise Valk [14,15,16], Geraint Rees [17], Sarah J. Tabrizi [1,19] & Peter McColgan[1,18,19]

Hyperconnectivity in functional brain networks occurs decades before disease onset in Huntington's disease. However, the biological mechanisms remain unknown. We investigate connectivity in Huntington's disease using Morphometric INverse Divergence (MIND) in three Huntington's disease cohorts (N = 512) spanning from two decades before the onset of symptoms through to functional decline. Here, we identify stage-specific profiles, with hyperconnectivity 22 years from predicted motor onset, progressing to hypoconnectivity through the late premanifest and manifest stages, showing that hypoconnectivity is correlated with neurofilament light concentrations. To understand the biological mechanisms, we investigate associations with cortical organization principles including disease epicentres and cell-autonomous systems, in particular neurotransmitter distribution. The contribution from disease epicentres is limited to late premanifest while cell-autonomous associations are demonstrated across the Huntington's disease lifespan. Specific relationships to cholinergic and serotoninergic systems localized to granular and infragranular cortical layers are identified, consistent with serotoninergic layer 5a neuronal vulnerability previously identified in post-mortem brains.

Huntington's disease (HD) is an autosomal dominant neurodegenerative disorder characterized by a combination of motor, cognitive and psychiatric symptoms[1]. Clinical diagnosis is defined by the existence of unequivocal motor symptoms, with people with HD (pwHD) being traditionally classified as premanifest (preHD) or manifest (mHD) based solely on this clinical milestone. The disease is caused by an expansion of the cytosine-adenine-guanine (CAG) trinucleotide repeat in the first exon of the *HTT* gene, encoding for the mutant Huntingtin protein (mHTT)[1] resulting in marked neuronal loss affecting medium spiny neurons in the striatum[2]. Simultaneously, there is widespread cortical atrophy also starting long before the onset of motor symptoms[3,4].

Increased functional connectivity is one the earliest features of HD occurring since childhood[5]. However, the underlying biological processes are unclear. Hyperconnectivity may represent a compensatory mechanism to preserve behavior in the presence of brain atrophy[6], although the correlation with plasma neurofilament light (NfL), a marker of axonal loss, suggests a pathological process[7]. Over time, increased connectivity turns into hypoconnectivity as motor symptoms emerge[8].

Organizational principles in the healthy brain can be used to understand multiscale systems-level disease mechanisms in neurodegeneration[9–14] where regions affected at later stages during

the course of the disease are connected with areas that degenerated first[15]. Healthy white matter networks in controls without brain disease overlap with atrophy patterns across psychiatric and neurodegenerative diseases such as schizophrenia and Alzheimer's disease, with lesions concentrated in hub regions[9,16]. Similarly, disease-specific epicentres estimated from task-free functional MRI data provide systems-level evidence for transneuronal spread of pathogenic proteins frontotemporal dementia and cortico-basal syndrome[17]. In pwHD, a network diffusion model[12] suggests that proximity is a key factor in the spread of pathology across adjacent neurons[8].

Genetic topography, which refers to the regional distribution of gene expression in the brain, has also been used to inform disease mechanisms in HD. We have previously revealed that brain regions showing cortico-striatal white matter loss are enriched for synaptic and metabolic genes in preHD individuals 15 years from onset[18]. In contrast, in individuals 25 years from disease onset, fMRI hyperconnectivity is specifically associated with neuronal gene expression[7].

Chemoarchitecture is an organizational principle that describes the regional distribution of neurotransmitter systems in the brain, and is related to atrophy in psychiatric diseases[19]. This organizational principle is of particular interest in HD given that cortical cell loss is prominent in serotoninergic neurons from infragranular layers, pointing towards cell-autonomous mechanisms (i.e., processes within a cell regulated by its own genes and biological features, independently of other cells) being associated with neuronal death[20–23].

Cell-autonomous mechanisms influence the pattern of neuronal loss in HD. Oligomeric forms of the mHTT protein are possibly toxic, leading to neurodegeneration in HD. Moreover, the mHTT protein sequesters transcription factors, resulting in transcriptional dysregulation, another core pathogenic mechanism in HD[24]. Similarly, mHTT alters mitochondrial dynamics interfering with ATP production and calcium buffering[25].

There is also extensive evidence of non-cell-autonomous mechanisms such as abnormal connectivity resulting in neuronal degeneration. Brain Derived Neurotrophic Factor is synthesized in cortical neurons but not in medium spiny neurons, and needs to be transported to the striatum[26]. Cortico-striatal degeneration disrupts this process facilitating striatal atrophy[27]. Also, alterations in non-neuronal cells can result in neuronal dysfunction and death. Expression of mHTT in astrocytes leads to motor deficits, and astrocyte pathology can exacerbate neuronal dysfunction through alterations in glutamatergic mechanisms[28]. Similarly, the expression of mHTT in microglia induces neuronal death through non-cell-autonomous mechanisms[29].

However, the relative contribution of cell-intrinsic mechanisms and non-cell-autonomous mechanisms is possibly specific to disease and stage, and both approaches have not been evaluated thoroughly -or simultaneously- in HD human brains[30].

Traditionally, brain structure has been investigated through univariate analyses of isolated parameters such as fractional anisotropy or sulcal depth in specific areas. However, brain regions do not function in isolation but form complex networks both at the cellular level and at the whole-brain scale. Understanding large-scale network architecture is therefore essential. Morphometric similarity mapping is a method used to develop structural morphometric similarity networks (MSNs) leveraging information from different micro- and macrostructural parameters through representing each brain region as a vector, followed by the pairwise correlations across brain regions. MSNs are associated with gene expression, recapitulate cortical cytoarchitectonic areas and are also related to axonal connectivity[31]. Alterations in MSNs are present in schizophrenic individuals as well as in people with Alzheimer's disease[32]. However, MSNs depend on summary statistics for each brain region instead of vertex-level data, and use Z-scores, constraining variability across brain regions[33].

The Morphometric INverse Divergence (MIND) technique is a method to investigate global brain connectivity that also leverages multiple imaging parameters at the vertex-wise level. In MIND, the similarity between brain regions is estimated through the Kullback–Leibler divergence between their distributions. The MIND method has been successfully applied to different cohorts[34–36] being more consistent with principles of cortical organization than other methods such as MSNs. In healthy brains, highly connected regions have similar transcriptional profile[11]. A gene co-expression network derived from the Allen Human Brain Atlas[11] showed a stronger correlation with MIND-derived connectomes compared to diffusion tensor imaging or MSNs. In addition, MIND-derived connectivity metrics were more closely associated with twin-based and SNP-based heritabilities than other imaging phenotypes, particularly in primary sensory regions. Brain networks estimated through MIND are correlated with axonal connectivity and cortical cytoarchitecture, being also more sensitive to predict age than similar techniques through machine learning models. Importantly, MIND can be used with imaging features obtained from simple T1-weighted MRI sequences such as cortical thickness, surface area, mean curvature, sulcal depth and gray matter volume. These characteristics support the use of MIND to investigate connectivity across the time course of in HD[33].

Here we had two objectives: First, to characterize connectivity across the disease course of HD, and its relationship with neuroaxonal damage. We hypothesized the existence of hyperconnectivity in HD participants 22 years from disease onset, progressing to hypoconnectivity through later stages. Second, we sought to understand the systems-level disease mechanisms driving this connectivity gradient, hypothesizing larger contributions from cell-autonomous mechanisms, as opposed to non-cell-autonomous systems.

To address the first objective, we utilized the MIND method[33] in some of the largest imaging cohorts in HD including the HD Young Adult Study (HD-YAS), with participants on average 22 years from disease onset[37], TrackOn-HD, 5 years from disease onset[38] and Track-HD, with mHD participants[39] (Fig. 1). To address the second objective, we explored the relationship between cortical networks throughout the different stages of HD, and a range of organizational principles including epicentres derived from healthy human brain fMRI and white matter networks[40] as well as further structural, functional and cell-autonomous systems.

In this work, we demonstrate a connectivity gradient across the disease time course of HD, with hyperconnectivity occurring 22 years from disease onset, followed by hypoconnectivity. The connectivity gradient in HD is markedly associated with the pattern of gene expression and neurotransmitter distribution in the healthy human brain, but there is only a limited association with healthy structural systems, predominantly in the late preHD stage. This indicates that cell-autonomous mechanisms have a marked impact on HD connectivity changes, while transneuronal spread of pathogenic factors plays a significant role exclusively during the period of accelerated neuronal loss present shortly before symptomatic onset[3]. Chemoarchitecture was specifically associated with serotonin receptors from infragranular layers, consistent with the selective vulnerability of neurons expressing the serotonin $5HT_{2C}$ receptor in post-mortem HD brains[20].

## Results

### Study population and MRI data

512 participants were included, comprising 252 pwHD and 260 controls. These data were obtained from three different studies: HD-YAS (57 early preHD and 60 controls), TrackOn-HD (85 late preHD and 89 controls) and Track HD (110 mHD and 111 controls) (Fig. 1a). Controls were age- and sex- matched with pwHD within each cohort. PreHD individuals were required to have a Total motor score (TMS) ≤ 5 and

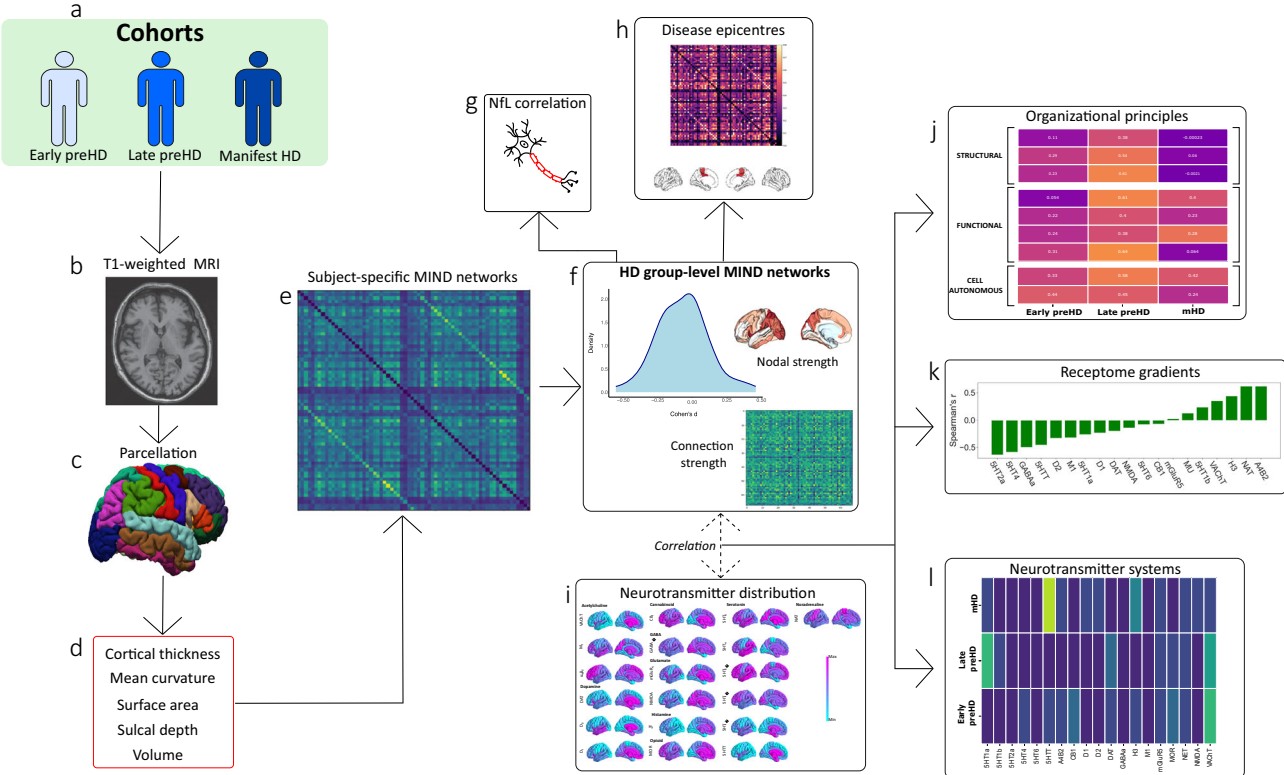

**Fig. 1 | Summary of analysis pipeline and main results.** Data from three cohorts (**a**) were used in this study: HD-YAS (57 early preHD, 60 controls), TrackOn-HD (85 late preHD, 89 controls) and Track-HD (110 manifest HD and 111 controls). Structural T1-weighted brain MRI scans (**b**) were parcellated using the 68 cortical region Desikan-Killiany[51] atlas with FreeSurfer's version 7 *recon-all* command (**c**). Cortical thickness, mean curvature, surface area, sulcal depth and gray matter volume were obtained (**d**) to compute subject-specific MIND networks. A linear mixed model was applied to each cohort obtaining z-scored effect sizes (Cohen's *d*) in patient populations versus healthy controls for nodal strength and connection strengths across MIND cortical similarity matrices (**e**). There was hyperconnectivity in HD participants two decades before motor onset, progressing to hypoconnectivity during later stages (**f**), the latter being associated with neuroaxonal damage (**g**). Healthy structural and functional connectomes were spatially correlated with nodal

strength to investigate the presence of disease epicentres (**h**). The distribution of neurotransmitter receptors and transporters was computed using PET data from more than 1200 healthy individuals using data from Hansen et al. (2022) (**i**). The relative contribution of additional structural, functional and cell-autonomous organizational principles was explored (**j**), finding that cell-autonomous gene expression and neurotransmitter distribution are major mechanisms contributing to nodal strength. Receptome analysis (**k**) showed that the first component of the distribution of neurotransmitter systems was associated with nodal strength. Finally, a neurotransmitter receptor analysis was performed disclosing a predominant association between the distribution of cholinergic and serotoninergic systems with nodal strength (**l**). The association with neurotransmitter systems was more prominent in granular and infragranular cortical layers. HD Huntington's disease, MIND Morphometric Inverse Divergence, preHD premanifest HD.

---

mHD individuals were required to score >5 in the same scale. See Table S1 for participant demographics.

HD-YAS was a single center study carried out in the London site with a Siemens scanner. In contrast, Track-HD and TrackOn-HD included participants from four study sites (Leiden, London, Paris and Vancouver). In Track-HD and TrackOn-HD 3T-MRI sequences were acquired in two scanner systems: Philips in Leiden and Vancouver and Siemens in London and Paris. T1-weighted data were acquired using a 3D Magnetization prepared rapid gradient echo (MPRAGE) sequence were in all studies.

ComBat harmonization, an empirical Bayesian method equivalent to batch effect correction which removes systematic differences that are caused by different sites or scanners[41] did not reveal significant differences in any brain region (Fig. S1, Table S2). Therefore raw data was used in this study. Site was added as a covariate in all analyses.

For further details please see the Methods section (MRI data acquisition and MRI data processing subsections)

**Hyper- and hypoconnectivity across HD disease time course**
In brain connectivity, the term 'node' refers to a brain regions. The brain 'connectome' is composed of all brain regions (nodes) and the connections between them in refs. 42,43.

Here, T1-weighted MRI-derived structural features parcellated through FreeSurfer's version 7[44] *recon-all* command were processed using the MIND approach[33] (Fig. 1b–e) to generate a connection measure of inter-regional similarity between two cortical areas across brain regions, producing symmetrical connectivity matrices for each participant in Track-HD, TrackOn-HD and HD-YAS. In these matrices each column and row represent a brain region of interest, termed a 'node', and each value in the matrix represents the 'connection' between these two brain regions. 'Node strength' represents the sum of the connection strengths for each brain region, reflecting how connected a specific area is. 'Connection strength' represents the magnitude of the connection between two brain regions. Robustness analyses were performed to examine the impact of ComBat harmonization as well as with more granular parcellation resolutions with the Schaeffer 100, 200, and 400 atlases[45] in the mHD cohort (Figs. S1–S2).

For node strength and connection strength, HD-specific MIND connectivity alterations were estimated through adjusted Cohen's *d*, providing a numeric value for the node strength and connection strength difference between pwHD and controls from each cohort (HD-YAS -early preHD-, TrackOn-HD -late preHD- and Track-HD -mHD-) in each node and in each connection (Figs. 1f, 2a–c).

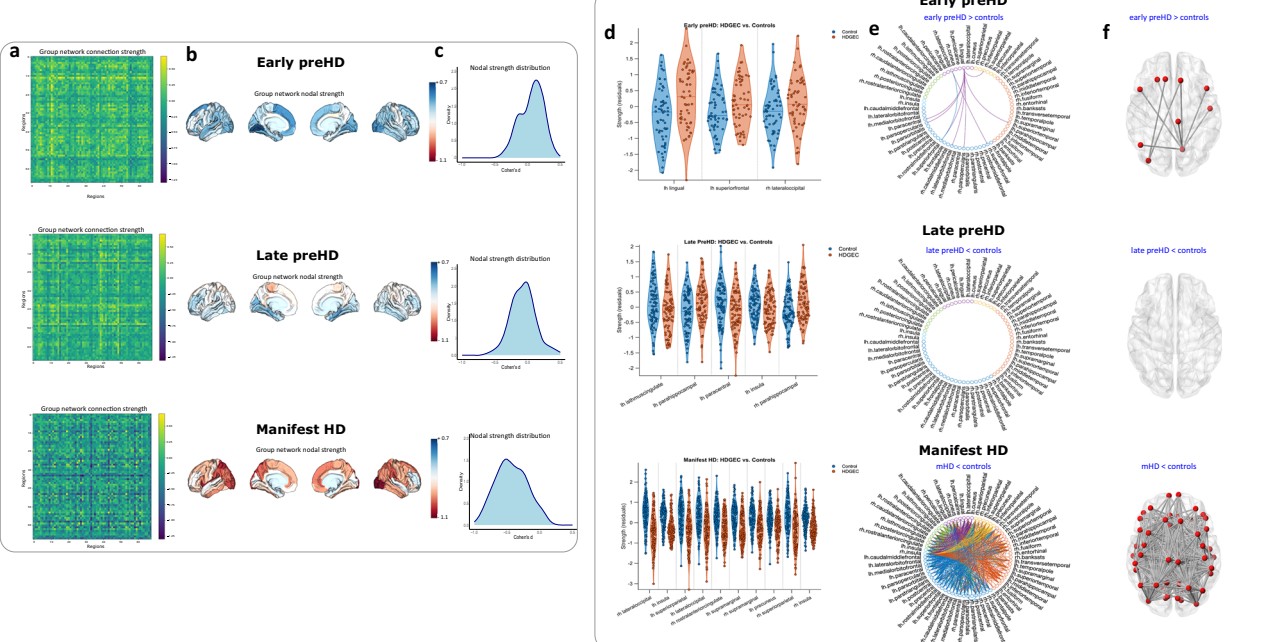

**Fig. 2 | MIND connectivity across the HD time course. a** Group-level cortical similarity matrices, each column and row represents a brain region of interest, termed a 'node' and each value in the matrix represents the 'connection' between two brain regions. The node strength represents the sum of all values for a given brain region, whereas the connection strength represents the magnitude of the connection between two brain regions. **b** Nodal strength for the differences between pwHD and controls across ROIs. In early preHD there are increases in connectivity predominantly occipital and frontal areas while in late preHD these differences attenuate. Finally, in mHD participants there are clear decreases in connectivity affecting also occipital and frontal areas. **c** Density plots representing the differences between pwHD and controls. Positive values in early preHD indicate increases in connectivity in pwHD compared with controls. These differences are predominantly negative in late preHD although most observations are around the null value. Finally, mHD participants have evident decreases in connectivity compared to controls. **d** Violin plots depicting ROIs showing significant differences at the uncorrected level in early preHD and late preHD. In mHD, for visualization

purposes, only the top five ROIs with larger effect sizes were shown, all of them significant following non-parametric permutation two-tailed tests, FDR correction at a $P < 0.05$. **e** Circular graphs depicting significant networks (network-based statistics, non-parametric permutation two-tailed t-tests FDR correction at a $P < 0.05$) in early preHD (early preHD > controls), late preHD (no significant results) and mHD (mHD < controls), five connections in the occipital lobe were significantly higher in mHD compared to controls and are represented in Fig. S3. **f** Axial view showing significant connections, (network-based statistics, non-parametric permutation two-tailed t-tests FDR correction at a $P < 0.05$) spheres indicate brain regions; lines indicate connections significantly different from controls. Source data are provided as a Source Data file. FDR false discovery rate, HD Huntington's disease, HDGEC Huntington's disease gene expansion carrier, mHD manifest Huntington's disease, MIND morphometric inverse divergence, NBS network-based statistic, pwHD people with Huntington's disease, preHD premanifest Huntington's disease, ROI region of interest.

Group-wise differences between pwHD and controls were explored for both connection strength and node strength, across cohorts. For connection strength and correlations with NfL concentrations (Fig. 1g), network-based statistics (NBS) were used (Fig. 2d–f). For node strength, non-parametric permutation two-tailed t-tests were used with false discovery rate correction (FDR) at a $P < 0.05$ across brain regions, to correct for multiple comparisons. Partial correlations were also performed between nodal strength and NfL in pwHD from each cohort.

For connection strength (Fig. 2, Tables S3–S5) in early preHD, 21.67 years from disease onset, NBS analyses revealed significant increases in connectivity relative to healthy controls ($P_{FWE} = 0.029$); no decreases in connectivity were observed. For late preHD we did not find significant increases or decreases in connectivity. For mHD, we observed widespread reductions in connectivity relative to healthy controls ($P_{FWE} < 0.0006$). Increased connectivity was also seen in five connections in mHD ($P_{FWE} = 0.022$), four occipito-occipital and one occipito-parietal connection (Fig. S3). Significant negative associations were observed between NfL and connection strength (Fig. 3, Tables S6–S7) in late preHD ($P_{FWE} = 0.044$) and mHD ($P_{FWE} < 0.0001$).

These results demonstrate hyperconnectivity in early preHD, progressing to widespread hypoconnectivity in mHD, where the negative relationship with NfL suggests hypoconnectivity has a pathological basis. In contrast, the increased occipital connectivity in

mHD did not correlate with NfL. It remains unclear if the hyperconnectivity observed in early preHD is pathological, compensatory or a combination of both.

Results at the node level were consistent with those observed at the connection level. In early preHD and in late preHD, there were no significant increases or decreases in node strength relative to healthy controls after FDR correction. For mHD, significant FDR-corrected reductions in connectivity compared to healthy controls were observed in 48 out of 68 brain regions ($P_{FDR} < 0.05$). No increases in connectivity were observed (Tables S8–S10). There were no correlations between NfL and node strength in early or late preHD after FDR correction. For mHD, significant FDR-corrected negative correlations with NfL were observed in 24 out 68 brain regions ($P_{FDR} < 0.05$) (Table S11).

The regional distribution of differences (i.e., spatial pattern) in MIND node strength was significantly correlated (Fig. S4, Tables S12–S13) between early preHD and late preHD ($rho = 0.26$, $P_{FDR} = 0.046$). There was no correlation between node strength in early preHD and mHD but the spatial pattern of node strength was also significantly correlated between late preHD and mHD ($rho = 0.38$, $P_{FDR} = 0.039$). These findings suggest that the areas with increased connectivity during very early stages of the disease are similar to brain regions that then experience connectivity loss later in the disease course.

# Neurofilament light protein and connectivity

Correlation coefficients

NBS analysis - Significant connections

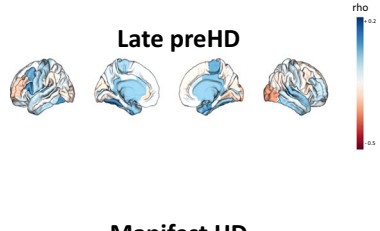

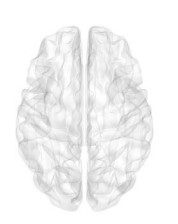

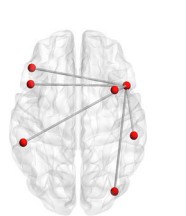

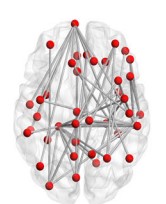

**Fig. 3 | MIND connectivity and plasma NfL correlations. a** There are positive correlations between plasma NfL and nodal strength in early preHD. During late preHD stages the direction of the correlation coefficients starts to change being again clearly negative in mHD stages. **b** Circular graphs showing significant associations (network-based statistics, non-parametric permutation two-tailed tests FDR correction at a $P < 0.05$) between connectivity and plasma NfL in early HD (no significant results), late preHD, and mHD. **c** Axial view showing significant

connections (network-based statistics, non-parametric permutation two-tailed t-tests FDR correction at a $P < 0.05$), spheres indicate brain regions; lines indicate connections significantly correlated with plasma NfL. Source data are provided as a Source Data file. HD Huntington's disease, mHD manifest Huntington's disease, MIND morphometric inverse divergence, NfL neurofilament light protein, NBS network-based statistics, preHD premanifest Huntington's disease, ROI region of interest.

## Disease epicentres are only associated with connectivity in late preHD

Disease "epicentres" are brain areas where there is a significant overlap between the connectivity pattern in healthy controls and disease-specific connectivity alterations. The nodal stress hypothesis suggests that highly connected areas are affected first in several neurodegenerative diseases. In addition, the structural and functional connectivity patterns of specific hub areas have been proposed to align with disease-associated atrophy acting as epicentres. The presence of these hubs suggest that either transneuronal spread of pathogenic factors or other mediators such as synaptic dysfunction, contribute to neural death in neurodegenerative diseases such as frontotemporal dementia or Alzheimer's disease[17]. To investigate the presence of disease epicentres in HD (Fig. 1h), HD-specific nodal strength data from the previous section was correlated with structural and functional connectivity matrices from healthy controls[40] generated from the Human Connectome Project (HCP) dataset[46] (28.73 ± 3.73 years).

HCP resting-state fMRI and structural data underwent minimal preprocessing as described in Glasser et al.[47] and was parcellated using the Desikan-Killiany 68-region cortical atlas. See the Methods section, subsection "Epicentre analysis" for additional information. Normative functional connectivity matrices were created computing the pairwise correlations between the time series in fMRI signal between brain regions. Normative structural connectivity matrices were determined through tractography-derived streamlines between regions. Since brain maps are spatially autocorrelated, non-parametric spin tests were used to generate null distributions that preserve spatial auto-correlation with 1000 random permutations, followed by FDR correction with 66 degrees of freedom. A brain region was defined as an epicentre in HD if nodal strength was significantly associated with the connectivity pattern in healthy controls.

In early preHD, no disease epicentres were identified using cortico-cortical or subcortico-cortical structural or functional

connectivity matrices. In late preHD, the paracentral and posterior cingulate areas showed significant associations between cortico-cortical structural connectivity and nodal strength (right paracentral, $rho = -0.28$, $P_{spin} = 0.034$; left paracentral, $rho = -0.24$, $P_{spin} = 0.034$; right posterior cingulate, $rho = -0.27$, $P_{spin} = 0.034$; left posterior cingulate, $rho = -0.30$, $P_{spin} = 0.034$). No additional disease epicentres were identified in late preHD or in mHD participants (Fig. 4).

These results suggest that the relative contribution of axonal connections varies during the course of the disease, being particularly prominent during late preHD stages, when potentially pathogenic factors such as the misfolded mHTT could spread along axons in highly connected regions. However, additional mechanisms such as synaptic dysfunction leading to aberrant signaling and eventual degeneration might also be involved, either independently or alongside transmission of pathogenic factors along axons. These phenomena are region-specific, targeting paracentral and cingulate cortices, which are known to develop early cortical gray matter loss in HD[3].

## Cell-autonomous mechanisms underlie connectivity loss in HD

The results outlined in the preceding section suggest a limited mechanistic role for axonal connections in driving connectivity loss across the HD disease time course. The next step was to explore multiple organizational principles of the healthy human brain in parallel, spanning structural, functional and cell-autonomous systems to identify additional biological mechanisms that account for these connectivity changes.

We explored associations with different organizational principles, obtained from datasets of healthy controls different from the HD-YAS, TrackOn-HD and Track-HD cohorts (Fig. 1j). First, with three structural organizational principles: laminar organization, white matter connectivity and Euclidean distance. For each principle, a brain region by brain region matrix was created with each value in the matrix representing the similarity between two regions. These matrices were then spatially correlated with node strength, representing the HD-specific

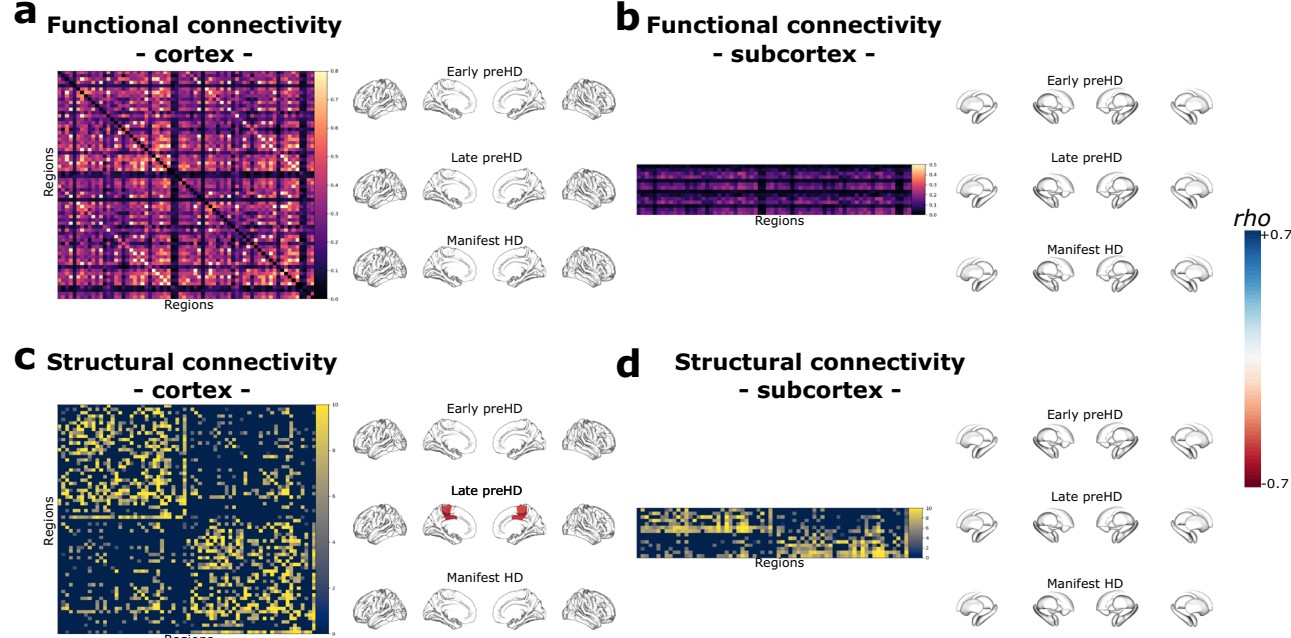

**Fig. 4 | Disease epicentre mapping.** Cortico-cortical functional (**a**) and structural (**c**) connectivity matrices in healthy controls. The colored areas in the brain surfaces depict regions (epicentres) whose connectivity profiles are significantly correlated with MIND nodal strength in each HD cohort. Only the paracentral and posterior cingulate gyri in late preHD were disease epicentres in the structural connectivity analysis (non-parametric spin permutation test, two-tailed, FDR correction at a $P < 0.05$, DF = 66). In (**b**, **d**) the matrices represent the subcortico-cortical functional (**b**) and structural (**d**) connectivity profiles, not being significantly associated with connectivity. Statistical significance was assessed through FDR-corrected spin permutation tests. Source data are provided as a Source Data file. DF degrees if freedom, HD Huntington's disease, FDR false discovery rate, MIND morphometric inverse divergence, preHD premanifest Huntington's disease.

connectivity differences in each cohort. All results were adjusted for spatial autocorrelation using FDR-corrected $P_{spin}$ tests, two-tailed, with 66 degrees of freedom (Fig. 5, Table S14).

In early preHD, late preHD and mHD no significant associations were seen between node strength, laminar organization or Euclidean distance (Table S14). In contrast to the anatomical region-specific relationships seen with disease epicentres and late preHD, these results suggest that these structural organizational principles have a minimal impact in connectivity changes at the whole-brain level, across the disease time course of HD.

We next explored associations with four different principles of functional organization obtained through neuroimaging techniques investigating the between-region similarities in fluctuations in neuronal activity across cortical areas. The four different ways of defining functional organization used either FDG-PET, resting state fMRI (rs-fMRI), dynamic rs-fMRI or magnetoencephalography (MEG). Significant associations were seen in late preHD ($rho = 0.61$, $P_{spin} = 0.016$), and mHD ($rho = 0.40$, $P_{spin} = 0.016$) for FDG-PET. No significant relationships were seen for early preHD or for fMRI network organizational principles in any cohort. These results suggest late preHD and mHD connectivity changes relate to FDG-PET brain networks, consistent with known altered FDG-PET cortical metabolism in preHD and mHD individuals[48,49].

Finally, we explored the associations between node strength and two cell-autonomous organizational principles: gene expression and neurotransmitter systems. In early preHD, significant associations were seen for neurotransmitter systems ($rho = 0.44$, $P_{spin} = 0.047$) but not gene expression. In late preHD, significant associations were seen for gene expression ($rho = 0.58$, $P_{spin} = 0.031$) but not neurotransmitter systems. Similarly, in mHD a significant association was seen for gene expression ($rho = 0.42$, $P_{spin} = 0.031$) but not neurotransmitter systems. These findings show that cell-autonomous organizational principles have a substantial impact on connectivity across the disease course, with neurotransmitter systems having a greater association in early HD and gene expression in late preHD and mHD.

In summary, cell-autonomous mechanisms are associated with connectivity alterations across the spectrum of HD. Although their relative contribution varies as the pathological load increases, these factors relate substantially to pathology during early stages of the disease and after the emergence of functional decline. Our findings suggest that the pathogenic process is initiated in susceptible neural populations decades before disease onset. Shortly before motor onset, axonal spread of pathogenic factors or other biological events contribute to accelerated atrophy. Eventually, in mHD patients, once there is substantial damage to axonal bundles, the degeneration depends again predominantly on cell-autonomous mechanisms.

**Receptome gradients across the disease spectrum**

The next step was to identify the specific aspects of these cell-autonomous mechanisms that account for the observed pathology. We have previously examined in detail the gene expression correlates of cortical degeneration in HD[50]. Therefore, we sought to understand the neurotransmitter systems that contribute to connectivity loss. PET neurotransmitter data for 19 receptors and transporters from more than 1200 healthy controls was parcellated using the Desikan-Killiany atlas[51] obtaining a 68-brain region by 19-neurotransmitter matrix[19] and compared with HD-specific nodal strength (Fig. 1k). Included neurotransmitters and transporters were Serotonin (5-HT$_{1A}$, 5-HT$_{1B}$, 5-HT$_{2A}$, 5-HT$_4$, 5-HT$_6$, 5-HTT), dopamine (D$_1$, D$_2$, DAT), norepinephrine (NET), histamine (H$_3$)Acetylcholine ($\alpha_4\beta_2$, M$_1$, VAChT), cannabinoid (CB$_1$), opioid (MOR), glutamate (NMDA, mGluR$_5$) and GABA (GABA$_{A/BZ}$).

Spearman-rank correlations were calculated between brain regions across neurotransmitter systems generating a receptome matrix that represents how similar pairs of brain regions are in terms of the neurotransmitter systems they express. Next, a dimensionality reduction technique was used to derive gradients representing the

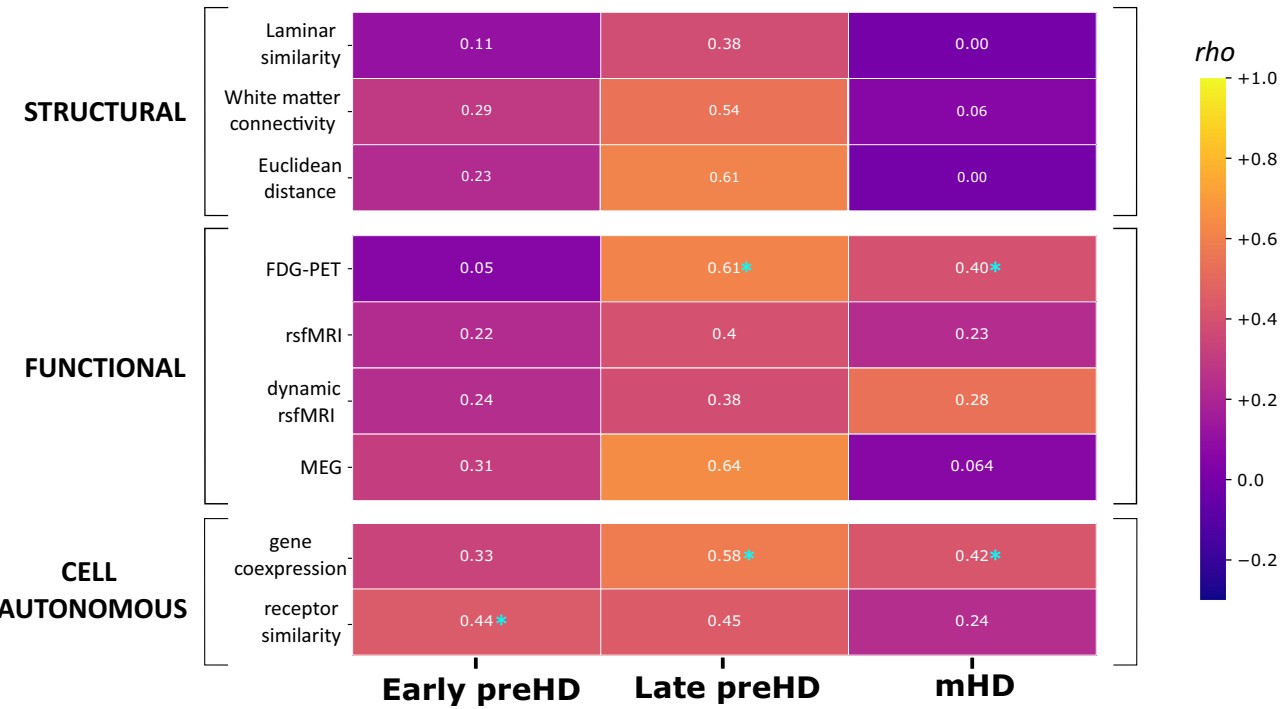

**Fig. 5 | Contributions of different organizational principles to MIND nodal strength.** Brain region by brain region matrices were constructed for each organizational principle using data from healthy controls. These were spatially correlated with nodal strength representing the differences in connectivity across the same regions between pwHD and controls (Cohen's *d*) in each cohort. Values and colors (yellow: higher; violet: lower) in each cell represent Spearman correlation coefficients. Statistical significance was assessed through non-parametric FDR-corrected spin permutation tests, two-tailed, DF = 66. Light blue asterisks indicate statistically significant results ($P_{spin} < 0.05$). Source data are provided as a Source Data file. DF degrees of freedom, FDG-PET [F[18]]-fluorodeoxyglucose positron emission tomography, FDR false discovery rate, MEG magnetoencephalography, MIND Morphometric Inverse Divergence, pwHD people with Huntington's disease, preHD premanifest Huntington's disease, rsfMRI resting-state functional MRI.

organizational principles in neurotransmitter density in healthy subjects.

The first gradient, describing 15% of the variance, loaded negatively on serotoninergic $5HT_{2A}$ and $5HT_4$ receptors, and positively on nicotinic receptor $\alpha_4\beta_2$ and on the noradrenaline transporter. The second gradient explained 14% of the variance, loading negatively on serotonin transporter and dopamine transporter and positively on nicotinic receptor $\alpha_4\beta_2$ and serotonin $5HT_{1B}$. Finally, the third gradient explained 13% of the variance and loaded positively on serotonin $5HT_{1A}$ and dopamine $D_2$ receptors and negatively on $GABA_A$ (Fig. 6).

We examined for correlations between these gradients spatially and node strength in each HD cohort, representing HD-specific differences in connectivity, using FDR-corrected spin tests, two-tailed, with 98 degrees of freedom. The first gradient was significantly correlated with node strength in early preHD participants (*rho* = −0.32, $P_{spin} = 0.001$) and in late preHD (*rho* = −0.53, $P_{spin} = 0.021$) but not with mHD (*rho* = −0.04, $P_{spin} = 0.911$). Gradients 2 and 3 did not reach statistical significance for any cohort (Fig. 6, Table S15).

These findings suggest that the cortical distribution of 5HTT, nicotinic receptor $\alpha_4\beta_2$ and noradrenaline receptor is associated with connectivity alterations in the early and late preHD stages.

**Serotoninergic and cholinergic systems drive HD connectivity**
To identify which specific neurotransmitter systems exert the greatest contribution to HD connectivity, a dominance analysis was performed[52]. This analysis assesses the variance of connectivity loss explained by each individual neurotransmitter receptor or transporter. A total of 19 neurotransmitter PET maps from healthy controls were investigated and the association with HD-specific nodal strength was computed through the $R^2_{adjusted}$ of each multilinear regression model

was tested using FDR-corrected spin tests, one-tailed, N = 68 regions (Fig. 1i, l).

Neurotransmitter systems were significantly associated with node strength, representing the HD-specific connectivity in each of the three HD cohorts: early preHD (total $R^2_{adj} = 0.36$, $P_{spin} = 0.019$); late preHD (total $R^2_{adj} = 0.62$, $P_{spin} = 0.019$) and mHD (total $R^2_{adj} = 0.37$, $P_{spin} = 0.019$) (Fig. 7).

The main top three neurotransmitter systems (Table S16) associated with node strength in early preHD participants were the vesicular acetylcholine transporter VAchT (% relative contribution to total $R^2_{adj} = 25.08\%$) followed by the cannabinoid receptor $CB_1$ (9.29%) and by μ-opioid receptors (9.01%). In late preHD the serotoninergic receptor $5\text{-}HT_{1A}$ (26.70%) predominated, followed by the vesicular acetylcholine transporter VAchT (21.07%) and the dopamine agonist transporter DAT (9.13%). Finally, in mHD the main association was with the serotoninergic transporter 5-HTT (36.44%) followed by the histaminergic receptor $H_3$ (13.11%) and the NMDA receptor (7.25%). Our findings again reflect the predominance of serotoninergic, cholinergic and dopaminergic systems influencing connectivity gradients across the spectrum of HD.

An autoradiography dataset including 15 neurotransmitter systems in 44 cortical areas from three postmortem brains without neurological disease was used to identify layer-specific associations with HD-specific connectivity changes (Tables S17–S19). Node strength in early preHD was significantly correlated with neurotransmitter distribution in the granular (total $R^2_{adj} = 0.61$, $P_{spin} = 0.010$) and infra-granular layers (total $R^2_{adj} = 0.54$, $P_{spin} = 0.021$). The main neurotransmitter systems were $5HT_2$ and $GABA_A$ both in the granular (% relative contribution to total $R^2_{adj}$, $5HT_2 = 30.01\%$, $GABA_A = 16.77\%$) and infragranular layers ($5HT = 27.51\%$, $GABA_A = 26.63\%$). In late preHD

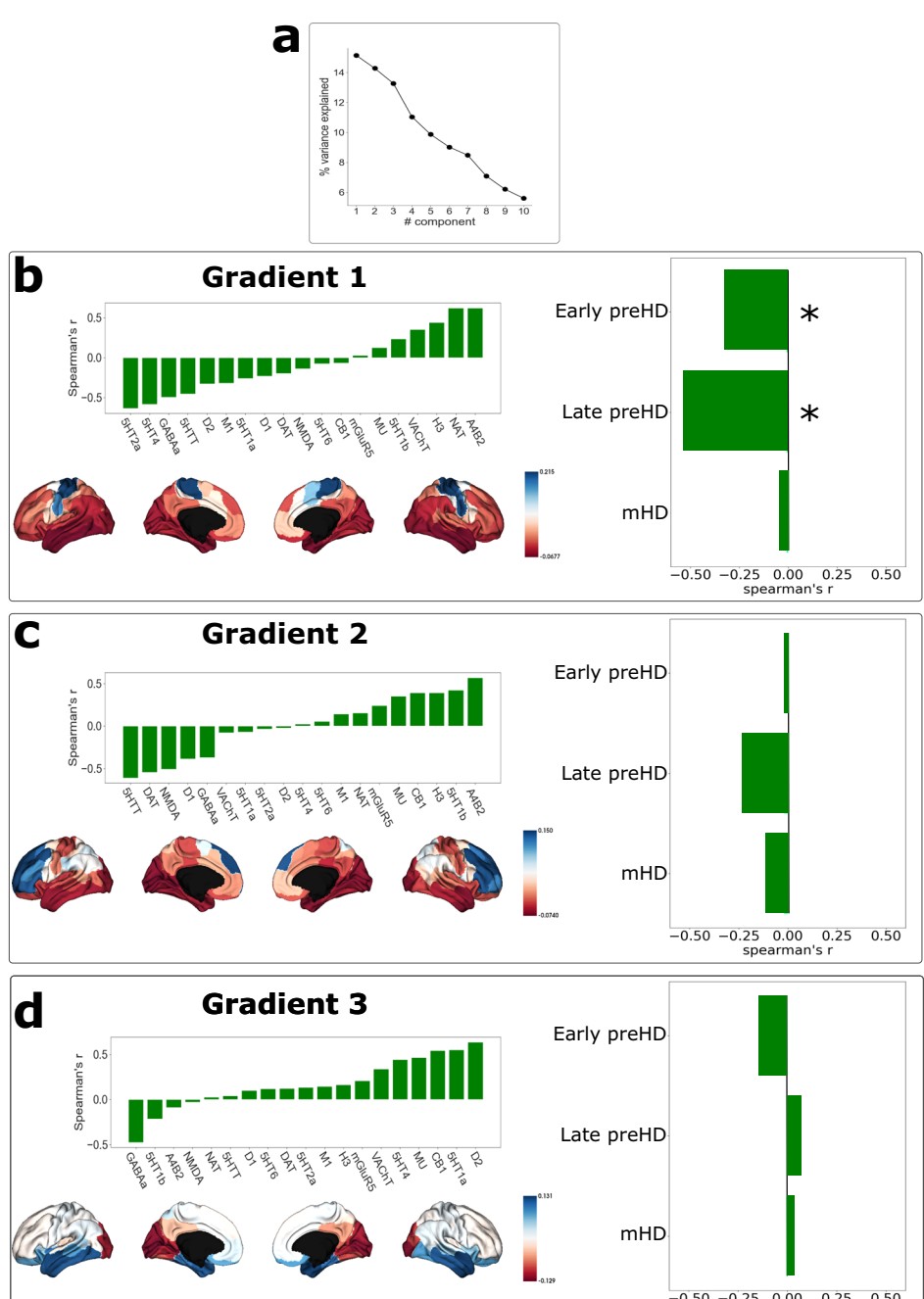

**Fig. 6 | Receptome gradients and MIND cortical similarity networks.** Neurotransmitter fingerprints were obtained using PET data from >1200 healthy controls to derive gradients of receptome organization. In (**a**) the proportion of variance explained by the different components following gradient decomposition is represented. Top panels in (**b**–**d**) show the Spearman rank correlations of cortical receptome gradients with individual neurotransmitter densities in healthy controls. The spatial distribution of these gradients is represented across ROIs in the bottom (**b**–**d**). The bar plots in the right side of (**b**–**d**) depict the Spearman rank correlations of receptome gradients with HD-specific node strength each cohort. The first gradient showed a strong significant correlation with connectivity particularly in late preHD and early preHD. Statistical significance was assessed through FDR-corrected spin permutation tests, two-tailed, DF = 98. Asterisks indicate significant results ($P_{spin} < 0.05$). Source data are provided as a Source Data file. DF degrees of freedom, FDR false discovery rate, HD Huntington's disease, mHD manifest Huntington's disease, preHD premanifest Huntington's disease.

and mHD there were no significant associations between connectivity and layer-specific neurotransmitter distribution.

In summary, our results indicate that neurotransmitter systems are robustly associated with connectivity loss across the HD disease time course, with acetylcholine exerting substantial influence in early and late preHD, while serotonin receptors are involved across disease stages. Moreover, layer-specific data concurs with previous evidence pointing towards early involvement of deep cortical layers early during HD pathogenesis, consistent with the early loss of cortico-striatal connections in HD[53,54], particularly in serotoninergic neurons from layer 5a[20].

## Discussion

Our first aim was to investigate connectivity changes across the time course of HD. Therefore, we leveraged high-quality data

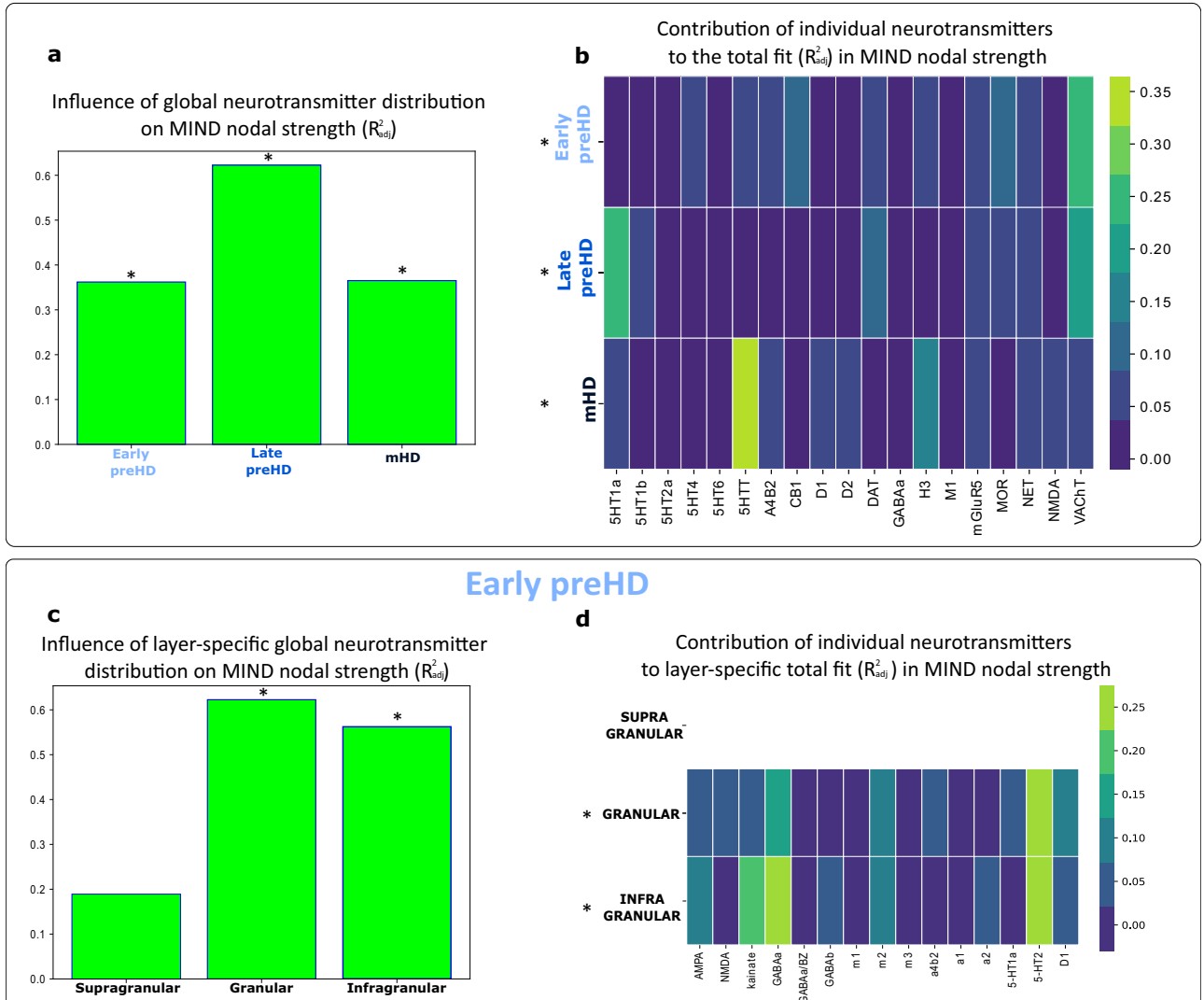

**Fig. 7 | Global and layer-specific neurotransmitter distribution and nodal strength for cortical connectivity networks in Huntington's disease.** A multi-linear model was used to determine the association between the distribution of the different neurotransmitter systems and node strength across the lifespan of HD. **a** illustrates how neurotransmitter system distributions map onto nodal strength across the different HD cohorts (total $R^2_{adj}$). Only significant results (*) assessed through FDR-corrected spin permutation tests, one-sided, $N = 68$ regions are represented in **a**, **b** (PET) and **c**, **d** (autoradiography). The dominance analysis in (**b**) distributes the fit of the model across each input variables showing the relative contribution of each variable. In (**c**), the total $R^2_{adj}$ for layer-specific auto-radiography results for nodal strength in early preHD are depicted, while (**d**), shows the relative contribution of each neurotransmitter system. Remaining auto-radiography analyses were not statistically significant in any cohort. Source data are provided as a Source Data file. FDR false discovery rate, HD Huntington's disease, mHD manifest Huntington's disease, MIND morphometric inverse divergence, preHD premanifest Huntington's disease. *$P_{spin} < 0.05$; PET positron emission tomography.

from three cohorts including 512 participants spanning more than two decades before disease onset until the emergence of functional decline. Hyperconnectivity was revealed as an early phenomenon in HD, being present more than two decades before the onset of clinical symptoms. As the disease advances, hyperconnectivity progresses to hypoconnectivity affecting similar areas, and being associated with neuroaxonal loss, as evidenced by the association with plasma NfL. Next, our second aim was to understand the biological factors underlying these connectivity changes. We show that the influence of disease epicentres in connectivity, and thus the possible transneuronal spread of pathogenic factors, is mainly limited to the period immediately before clinical motor onset. In contrast, cell-autonomous mechanisms, especially gene expression and neurotransmitters such as serotoninergic and cholinergic systems are robustly

associated with connectivity gradients along the course of HD, particularly long before symptom onset and after the emergence of motor symptoms.

Our results indicate that the relative weight of factors such as structural and functional systems-level organizational principles and cell-autonomous mechanisms have an impact in the HD brain that varies across disease stages. We hypothesize that initially cell-autonomous mechanisms predominate as the presence of the *HTT* mutation initiates the pathogenic process in vulnerable cells with similar biological features. This results in hyperconnectivity in cortical similarity networks. Close to clinical motor onset, cell-to-cell transmission of pathology along relatively intact axonal tracts becomes more relevant. This period, when there is a convergence between cell-autonomous and non-cell-autonomous[55,56] mechanisms corresponds to the acceleration of neuronal loss in perimanifest patients that has

been demonstrated through imaging[3] and biofluid[57] biomarkers. After the onset of symptoms and the emergence of functional decline, there is extensive axonal damage resulting in hypoconnectivity[58] potentially interfering with cell-to-cell spreading of pathogenic factors. During this period cell-autonomous mechanisms again predominate, with neurodegeneration becoming asynchronous. These findings are consistent with recent evidence of asynchronous somatic expansion of the CAG repeat in medium spiny neurons during early stages of degeneration, suggesting a similar mechanism in the cortex[59].

We identified hyperconnectivity in participants 22 years before the expected onset of symptoms, while connectivity decreases were detected five years before the onset of symptoms, becoming widespread during manifest stages. This initial hyperconnectivity is consistent with existing evidence and has previously been suggested to represent compensation[5,6,7]. Neuronal compensation has been proposed to be responsible for the preserved cognitive performance in the presence of neuronal loss in HD, resulting from increased activation within existing networks. Previous studies in preHD showed increased fMRI activation in the right parietal cortex during a working memory task[6]. This hyperactivation decreases as atrophy becomes more prominent, when individuals progress towards clinical motor onset[6]. Similarly, children with CAG repeats in the adult-onset range, have increased fMRI connectivity between the striatum and the cerebellum before the onset of symptoms[5], and there is experimental evidence of abnormal excitatory activity in HD mice leading to neuronal death, that can be prevented through early normalization of glutamatergic transmission[60]. Interestingly, we found that in mHD there were also focal increases in connectivity exclusively in occipital regions, not correlating with plasma NfL. However, these were limited to the occipital cortex. The occipital lobe is sparsely connected to the striatum relative to other brain areas[61]. Our previous work has shown that changes in the occipital cortex of pwHD are largely developmental rather than neurodegenerative[50] and occipital hyperconnectivity could be related to compensation, or different mechanisms governing regions densely connected to the striatum.

In the present study, we did not find significant correlations between connectivity with NfL in early preHD. However, the hypoconnectivity present during late preHD and manifest stages of HD was robustly associated with NfL. The absence of correlations with NfL in early preHD, leaves the possibility that the hyperconnectivity identified in preHD may be compensatory or pathological or potentially a combination of these processes across different brain regions.

There is extensive evidence showing that inter-hemispheric connections are affected early during the course of HD[18,62]. This may be related to the loss of cortical intratelencephalic neurons that project to the striatum and across the corpus callosum[53]. In keeping with this, the corpus callosum, the main conduit of information between cerebral hemispheres has shown early involvement with a temporally and topologically selective pattern of degeneration in HD both using structural[63] and diffusion MRI[64].

Next, we investigated the biological mechanisms driving connectivity gradients. We observed associations with cell-autonomous mechanisms across cohorts. However, although region-specific disease epicentres were identified, we did not find an association between global structural organizational principles with connectivity in early preHD and mHD participants, suggesting that only certain structural connections constrain connectivity changes in late preHD, while cell-autonomous mechanisms are crucial across disease stages. In contrast, in preHD participants closer to motor onset, exclusively the paracentral and posterior cingulate regions were associated with structural connectivity, acting as disease epicentres, and suggesting that transneuronal spread of pathogenic factors, is also involved in disease pathogenesis together with cell-autonomous mechanisms. This is in keeping with atrophy of the paracentral and posterior cingulate areas occurring early during the course of HD being associated with clinical dysfunction[3].

Interestingly, there was no association between subcortical epicentres in healthy controls and cortical MIND networks in HD. This suggests that degeneration of cortico-cortical connections may be either independent or temporally dissociated from subcortical pathology. Indeed the very early occurrence of striatal pathology, seen 24 years before disease onset in HD[37] suggests the latter. The absence of subcortical epicentres may also be related to methodological factors such as difficulty in the accurate segmentation of small subcortical nuclei using MRI methods.

We also observed significant correlations between the distribution of glucose metabolism and connectivity in late preHD and mHD. These findings concur with previous evidence of reductions in [18 F] FDG uptake in cortical areas in preHD and early mHD reflecting abnormal mitochondrial function and supporting the interaction between connectivity alterations and energy production in HD[65].

Our findings indicate that the relative weight of cell-autonomous and non-cell-autonomous mechanisms varies across the spectrum of HD. Cell-autonomous organizational principles were robustly associated with connectivity alterations across the stages of HD. There were significant correlations with gene expression in late preHD and mHD. The association between gene expression and neuronal death in HD has been extensively explored[66,67]. Interestingly, a recent study has shown that medium spiny neurons, but not other striatal cells, undergo somatic expansion, which subsequently triggers severe changes in gene expression in these neurons leading to neuronal death and atrophy[59,66]. In addition, cortical infragranular neurons have decreased expression of dendritic and synaptic genes and also demonstrate extensive somatic expansion[20].

We demonstrate significant associations between chemoarchitecture organization globally and connectivity in early preHD, while specific receptors show associations across cohorts, including serotoninergic and cholinergic receptors in line with previous evidence[65]. Our results suggest that serotoninergic receptors mediate hyperconnectivity during early stages of neurodegeneration in HD. The association between connectivity changes in early preHD participants far from clinical motor onset and $5HT_2$ distribution was specific to the granular and infragranular layers, while in late preHD and mHD participants, connectivity changes were associated with the distribution of the serotoninergic receptor $5HT_{1A}$ and serotoninergic transporter density across the entire cortical ribbon, indicating that these associations may be progressive. Similarly, a previous study examining post-mortem cortical tissue from HD donors found that somatic expansion mediates selective vulnerability of layer 5a serotoninergic pyramidal neurons expressing *HTR2C*, encoding for the excitatory serotoninergic $5HT_{2C}$ receptor[20]. For further discussion about additional neurotransmitter systems see Supplemental Note.

There are some limitations to our study. The new Huntington's disease Integrated Staging System (HD-ISS) characterizes individuals using objective information from biomarkers and clinical scores, rather than the traditional clinical classification into preHD and mHD[68]. In this work we used information from observational cohorts recruited before the HD-ISS was developed, but future studies should classify pwHD using the HD-ISS. Also, in our late preHD cohort, controls were significantly older than *HTT* expansion carriers. This is inherent to the age-dependent penetrance of HD in studies where controls are age-matched either with preHD or with mHD participants. However, all analyses were adjusted for age.

Data from different cohorts and sites were used in our study. This could potentially lead to variability. However, in Track-HD and TrackOn-HD the structural imaging acquisition protocols were identical, and very similar to HD-YAS. In addition, all images underwent rigorous quality control (QC) through a central, independent reader after acquisition, and image processing was consistent across cohorts. Moreover, ComBat harmonization[41] was performed without observing significant differences.

Here, we correlated the cortical distribution of neurotransmitters in healthy controls with HD-specific connectivity maps. Although there are marked differences in the striatal densities of neurotransmitters between HD patients, the distribution of neurotransmitter densities in the HD cortex is largely similar to that in healthy controls[65]. Moreover, post-mortem brains are scarce and usually reflect end-stages of HD rather than the preHD or early mHD individuals included in this work. We focused on the cortex rather than the striatum since there is extensive evidence suggesting cortical involvement beginning in the very early stages of HD[50] and the cortex is relatively understudied in HD relative to the striatum. Additionally, we only used structural metrics to obtain MIND networks, in line with the original MIND study. Potentially, DWI-metrics or fMRI indexes could be included. However, not all participants in our study had diffusion or functional imaging data, which would limit the interpretability of our findings. Moreover, the differences in acquisition parameters might have a potential impact in these results. Other large datasets with high quality T1-images following standardized acquisition techniques are available, but only smaller cohorts usually with heterogeneous acquisition parameters have other imaging sequences. Finally, the Desikan-Killiany[51] 68-region parcellation atlas was chosen to make our results comparable with the ENIGMA dataset, but our findings were robust when more granular parcellations were used.

In summary, we demonstrate a gradient of connectivity alterations across the HD disease time course which is driven by cell-autonomous mechanisms in early preHD, a combination of cell-autonomous, structural and functional systems-level mechanisms in late preHD and again predominantly cell-autonomous mechanisms in mHD. These cell-autonomous mechanisms specifically implicate serotonin receptors in granular and infragranular layers, consistent with recent findings from postmortem studies. Our results provide evidence for the biological mechanisms of cortical pathology across the time course of HD.

## Methods

### Ethics

Ethics approval for the TRACK-HD and TrackOn-HD studies was granted for each site by the following boards: The Clinical Research Ethics Board, University of British Columbia (Vancouver); Comites de protection de personnes, Hopital de la Pitie-Salpetriere (Paris); Medical Ethics Committee of the Leiden University Medical Center (Leiden) and the National Hospital for Neurology and Neurosurgery and Institute of Neurology Joint Research Ethics Committee (London). Ethics approval for the HD Young Adult Study was granted by the London Queen Square Research Ethics Committee

Written informed consent was obtained from each participant. Our research complies with all relevant ethical regulations.

### Participants

Data from three different cohorts were used in this study; Track-HD is a longitudinal multicentric study that included 366 participants aged 18–65 from four study sites (Leiden, London, Paris and Vancouver). Study participants were rigorously evaluated using a standardized protocol and a robust design with four annual visits between 2008 and 2011. Sex and/or gender of participants was determined based on self-report. Key inclusion criteria were: ability to tolerate MRI scans and sample donation, while exclusion criteria were moderate/advanced HD (TFC ≤ 6) at baseline, major psychiatric, neurological or medical illness, significant head or hand injuries, predictable non-compliance or active participation in a clinical trial. The study was initially composed of 120 preHD, 123 mHD, and 123 age- and sex-matched controls at baseline. Data from the baseline visit was used in this study. Participants were evaluated with a comprehensive battery that included cognitive, neuropsychiatric, the unified Huntington's disease rating scale (UHDRS) and quantitative motor scales, physical and neurological examination, medical and psychiatric history, 3 T brain MRI and blood samples. For more information see Tabrizi et al.[4]. We included only manifest patients and controls from Track-HD in the primary analysis. Fourteen participants did not have complete clinical data or did not have structural MRI while eight participants failed QC after image processing leaving a total of 221 participants (110 mHD and 111 controls).

TrackOn-HD was the extension study from Track-HD. It collected data between 2012 and 2014 and focused on late preHD participants evaluating the mechanisms to maintain normal function despite the presence of macrostructural atrophy. It followed 112 controls, 110 late preHD and 21 mHD participants with annual visits. Newly recruited premanifest patients were required to have a CAG repeat length of at least 40 and a DBS (DBS = Age * (CAG-35.5)) larger or equal to 250. Inclusion and exclusion criteria were similar in both studies, with the exception of recruiting predominantly late preHD participants for TrackOn-HD. Assessments were similar to Track-HD, but there was an emphasis on advanced imaging including fMRI and DWI sequences. For more information see Kloppel et al.[38]. In the present study, only late preHD participants were included from the TrackOn-HD cohort, using data from the baseline visit. Forty-two participants did not have complete clinical data or structural MRI while six participants failed QC after imaging processing, leaving a total of 174 participants (85 late preHD and 89 controls).

Since TrackOn-HD participants did not have available plasma NfL data, we performed the NfL correlation analysis with late preHD participants from Track-HD. Seventeen late preHD participants from Track-HD did not have complete clinical data or did not have structural MRI. The remaining 103 late preHD participants from Track-HD passed QC after imaging processing. There were no significant differences in age, gender distribution, site, DBS or CAG repeat length between late preHD participants from Track-HD and those from TrackOn-HD. For more detailed information see Table S20.

The HD-YAS study included 64 early preHD and 67 control participants matched for age, sex and education. Early preHD participants had a CAG repeat >39, DBS ≤ 240 and a UHDRS Total Motor Score (TMS) ≤5. Control participants were gene-negative family members or individuals with no familial history of HD. Participants were excluded if they presented recent drug or alcohol abuse and/or dependence, neurological or significant psychiatric co- morbidity, brain trauma or contraindication to MRI. All participants underwent an extensive battery of cognitive and neuropsychiatric testing, clinical and medical history, neuroimaging, blood sampling and optional CSF collection. For more information see Scahill et al.[37].

Eight participants did not have complete clinical data or structural MRI, while six additional participants failed QC after imaging processing, leaving a total of 117 participants (57 early preHD and 60 controls).

### MRI data acquisition

In Track-HD and TrackOn-HD 3T-MRI sequences were acquired in two scanner systems: Philips in Leiden and Vancouver and Siemens in London and Paris. All data were visually inspected by IXICO Ltd for QC purposes. Scanning protocols were standardized between sites and inter-scanner comparisons were performed using human volunteers or phantoms. T1-weighted data were acquired using a 3D MPRAGE sequence with a slice thickness of 1.0 mm. Acquisition parameters in Siemens scans were repetition time = 2200 ms, echo time = 2.2 ms, flip angle = 10°, field of view = $256 \times 256 \times 280$ mm$^3$. 208 slices without gap were acquired for each volume. In Philips, volumetric acquisition parameters were repetition time = 7.7 ms, echo time = 3.5 ms, flip angle = 8°, field of view = $224 \times 224 \times 240$ mm$^3$. 164 slices without gap were acquired for each volume. For more information see Tabrizi et al.[4] and Kloppel et al.[38].

In HD-YAS, MRI data were acquired on a 3 T Siemens PRISMA scanner. All data were visually inspected by the UCL team for QC purposes. T1-weighted images were acquired using a 3D MPRAGE sequence with a slice thickness of 1.0 mm. Acquisition parameters were repetition time = 2530 ms; echo time = 3.34 ms, flip angle = 7°, field of view = 256 × 256 × 176mm³. For more information see Scahill et al.[37].

## MRI data processing

Baseline 3D T1 volumetric images were bias-corrected using the N3 algorithm[69]. Next, FreeSurfer version 7 (http://surfer.nmr.mgh.harvard.edu) was run via the default *recon-all* pipeline. Cortical thickness, surface area, volume, mean curvature, and sulcal depth were extracted from the Desikan-Killiany 68-region[51] cortical atlas.

Quality control was performed before analysis and included visual checks of the cortical segmentations using the ENIGMA protocols (http://enigma.usc.edu/protocols/imaging-protocols). Histograms of all regions' values for each site were also computed for visual inspection.

Subjects with large errors in cortical segmentations were removed from the analysis, detailed in the previous sections.

## MIND connectivity analysis

We used the mesh reconstructions of the cortical surface generated from T1-weighted MRI scans using FreeSurfer's version 7 *recon-all* command. Each vertex was identified through five structural MRI features: Cortical thickness, surface area, volume, mean curvature, and sulcal depth. To assess the similarity among cortical areas, each MRI feature was z-scored across all vertices and then combined for each cortical area defined by a predefined parcellation template. This process yielded a regional multivariate distribution. Subsequently, a pairwise distance matrix was constructed utilizing a k-nearest neighbor density algorithm to compute the symmetrized Kullback–Leibler divergence, between every pair of regional multivariate distributions. Finally, the Kullback–Leibler divergence was mapped between each pair of regions to estimate the inter-areal MIND similarity, which ranges from 0 to 1. Higher values denote greater similarity between regions. For more information see ref. 33.

The values for each map were z-scored effect sizes (Cohen's *d*) for each ROI in each cohort in patient populations versus healthy controls using linear mixed models to estimate sample-wise nodal strength and connection strengths across cortical regions using age, sex, site and total intracranial volume as covariates. These covariates are in line with previous research in neuroimaging in HD[4].

NBS version 1.2 (https://sites.google.com/site/bctnet/comparison/nbs) was used to investigate group differences in connection strength between pwHDs and controls and associations with plasma NfL in each cohort. Here a test statistic is calculated for each connection independently. A primary threshold ($P < 0.05$, uncorrected) is then applied to form a set of suprathreshold connections. Permutation testing is then used to calculate a family-wise error (FWE) corrected P-value for each set of suprathreshold connections. For these analyses, permutation testing using unpaired *t*-tests and 5000 permutations, as per the default NBS options, was performed on a general linear model that included age, gender, site and total intracranial volume as covariates. A test statistic was then computed for each connection and a default threshold applied ($t = 3.1$) to produce a set of suprathreshold connections that displayed significant between-group connectivity differences. FWE-correction was applied at $P = 0.05$. Results were represented using circular graphs and using the BrainNet Viewer (https://www.nitrc.org/projects/bnv). For node strength non-parametric permutation (10,000 permutations) 2-tailed *t*-tests were used with false discovery rate correction (FDR) across brain regions, to correct for multiple comparisons. Age, sex, site and total intracranial volume were used as covariates in all analyses.

Pearson's correlations were performed between nodal strength for each brain region in each subject and plasma NfL, across each cohort. Statistical significance was tested using a two-tailed $\alpha = 0.05$ and FDR was used to correct for multiple comparisons using the Benjamini-Hochberg procedure.

Visual representation of cortical surfaces was performed with the 'plot_cortical' function of the ENIGMA toolbox (https://enigma-toolbox.readthedocs.io/)[70].

## Epicentre analysis

Axonal tracts facilitate the interaction between different brain regions. One suggested mechanism to explain the region-specific neuronal loss in neurodegeneration has been the spread of pathogenic factors across axonal connections and consequently, brain networks have been used to predict the atrophy patterns of neurodegenerative diseases[40].

Therefore, in our study, disease epicentre mapping was used to explore the influence of structural and functional connectivity profiles in the different brain regions on cortical similarity networks across the spectrum of HD. The presence of multiple disease epicentres would support that transneuronal spread of mHTT and/or other pathogenic factors.

Data already processed from the HCP dataset[46], including a group of healthy adults ($n = 207$, 83 males, age range 22 to 36 years) was used to generate normative cortico-cortical and subcortico-cortical structural and functional connectivity matrices as described in Larivière et al.[40]. Diffusion MRI data from the HCP dataset was intensity-normalized with the mean B0 image. Next, diffusion data was corrected for distortions caused by motion and eddy currents as well as for susceptibility artifacts.

Resting-state fMRI data from the HCP dataset underwent first distortion as well as subject-motion correction. Next, resting-state fMRI data was corrected for magnetic field bias, skull removed and intensity-normalized before being mapped to the MNI152 space. FMRIB's ICA-based X-noiseifier (FIX)[71] was used to automatically remove additional noise components related to presence of white matter, pulsation or movement. For additional information about preprocessing see Glasser et al.[47]. Finally, diffusion MRI and resting-state fMRI data from the HCP dataset were parcellated with the 68-cortical and 18-subcortical regions of the Desikan-Killiany atlas.

Normative structural connectivity matrices were generated using MRtrix3[72]. T1-weighted images were used to perform anatomically constrained tractography. Forty million streamlines were produced with a fractional anisotropy threshold of 0.06 and a maximum tract length of 250. The spherical-deconvolution informed filtering of tractogram (SIFT2) algorithm was then used to reconstruct whole-brain streamlines weighted by the cross-sectional multipliers. These were subsequently mapped onto the 68-cortical and 14 subcortical regions from the Desikan-Killiany atlas to generate subject specific structural connectivity matrices. A distance-dependent thresholding procedure was used to estimate group-averages followed by log transformation.

Normative functional connectivity matrices were generated computing the correlations between the time series of 68-cortical regions with all cortical and subcortical regions. Negative connections were zeroed. Group-averaged functional connectomes were computed through z-transformation of aggregated subject-specific connectivity matrices.

Next, spin permutation tests with 1000 permutations were performed to investigate the spatial correlation between the normative cortical and subcortical epicentre maps with whole-brain Cohen's *d* maps for MIND cortical similarity networks across all three cohorts. Resulting $P_{spin}$ values were corrected for multiple comparisons using the FDR Benjamini-Hochberg correction (Table S22). For more information see ref. 40.

## Organizational principles of the healthy brain

Axonal fiber bundles connect different neuronal populations. These can be traced in vivo through modeling of DWI data, obtaining

structural connectivity metrics which are associated with brain function in healthy controls, as well as with disease phenotypes in neurological conditions[73]. However, structural connections are not the only factor associated with the function of different brain areas. Instead, regions with similar molecular, cytoarchitectonic and electrophysiological architecture tend to share similar biological roles[74].

Therefore, we investigated the relationship between different biological factors in healthy brains and nodal strength across the time course of HD. Structural, functional and cell-autonomous organizational principles were obtained in healthy controls and correlated with HD-specific connectivity alterations using Cohen's $d$ for nodal strength as previously described.

We investigated three structural organizational principles: laminar organization, white matter connectivity and Euclidean distance; four functional organizational principles: FDG-PET, rs-fMRI dynamic fMRI and MEG and two cell-autonomous organizational principles: gene expression and neurotransmitter receptors[30].

For instance, a matrix was created with each value indicating the similarity between two areas based on their laminar organization, as defined by the histological BigBrain cell-staining atlas. The Louvain algorithm, which maximizes positive and negative edge strength was used to identify communities for each organizational principle in a manner that maximizes the function below.

Equation 1 Louvain Algorithm

$$Q(\gamma) = \frac{1}{m^+}\left[w_{ij}^+ - \gamma p_{ij}^+\right]\delta(\sigma_i, \sigma_j) - \frac{1}{m^+ + m^-}\sum_{ij}\left[w_{ij}^+ - \delta p_{ij}^-\right]\delta(\sigma_i, \sigma_j)$$

Here, $w_{ij}^+$ is the network of only positive correlations (and $w_{ij}^-$ is the equivalent for negative correlations. The null model is $p_{ij}^{\pm} = \left(s_i^{\pm}s_j^{\pm}\right)/(2m^{\pm})$ and the expected density of the connections between nodes i and j was $s_i^{\pm} = \sum_j w_{ij}^{\pm}$ and $m^{\pm} = \sum_{i,j>i} w_{ij}^{\pm}$. The community assignment of node $i$ is $\sigma_i$ and $\delta(\sigma_i, \sigma_j)$ is the Kronecker function, being equal to 1 when $\sigma_i = \sigma_j$ and 0 otherwise. $\gamma$, the resolution parameter scales the relative importance of the null model, making it easier ($\gamma > 1$) or harder ($\gamma < 1$) for the algorithm to uncover many communities. For more information see[30].

Next, the different measurements for each brain region were combined through similarity network fusion. Spin tests with 1000 permutations were used to adjust for spatial autocorrelation[75]. Resulting $P$ values were adjusted using the FDR Benjamini-Hochberg procedure (Table S23).

Nodal strength, estimated through the Cohen's $d$ for the differences between HD and controls parcellated using the 68-region Desikan-Killiany[51] atlas were used to estimate disease exposure, as previously explained. Here, connections with negative strength were assigned a strength of 0, disease exposure of a node $i$ was defined as:

Equation 2 Disease exposure of a node

$$D_i = \frac{1}{N_i}\sum_{j\neq i, j=1}^{N_i} d_j \times c_{ij}$$

Where $N_i$ was the number of positive connections made by region $i$, $d_j$ the Cohen's $d$ for MIND connectivity at region $j$, and $c_{ij}$ the connection strength between regions $i$ and $j$. For more information see ref. 30.

## Receptome gradients

PET neurotransmitter receptor and transporter data from healthy controls ($n = 1238$, 718 males) was parcellated with the 68-region Desikan-Killiany atlas[51] resulting in a z-scored 68 region × 19 neurotransmitter matrix[19]. Next, a parcel-by-parcel Spearman's correlation of neurotransmitter receptor densities was performed, resulting in a matrix of node-level interregional similarity in the distribution of neurotransmitter densities, the receptome. A normalized angle similarity kernel was used to generate an affinity matrix. Finally, diffusion embedding was used to obtain the main organizational axes of cortical chemoarchitecture.

The first gradient, describing 15% of the variance, was differentiated between somatomotor and the inferior occipital and temporal lobe regions. It loaded negatively on serotoninergic $5HT_{2A}$ and $5HT_4$ receptors, and positively on nicotinic receptor $\alpha_4\beta_2$ and on the noradrenaline transporter. The second gradient explained 14% of the variance and spanned from frontal to occipital regions and loaded negatively on serotonin transporter, dopamine transporter and positively on nicotinic receptor $\alpha_4\beta_2$ and serotonin $5HT_{1B}$. Finally, the third gradient explained 13% of the variance and delineated an axis from the temporal lobe to the occipital lobe, it loaded positively on serotonin $5HT_{1A}$ and dopamine $D_2$ receptors and negatively on $GABA_A$. For more information, see ref. 76.

In each cohort, Cohen's $d$ nodal strength MIND networks for the differences between pwHD and controls, parcellated with the 68-region Desikan-Killiany, was used to investigate the association between receptome gradients and MIND cortical similarity networks in HD. Statistical significance was investigated accounting for spatial autocorrelations using spin permutations with 1000 permutations to create randomly permuted brain maps. Results were subsequently corrected using the FDR Benjamini-Hochberg procedure (Table S24).

## PET neurotransmitter analysis

The z-scored parcellated PET data for 19 neurotransmitter systems described in the previous section was used to determine the association of each neurotransmitter system with MIND cortical similarity networks. To ascertain the relative contribution of each neurotransmitter, a dominance analysis was performed. Cohen's $d$ values for the differences between patients and controls in MIND cortical similarity networks in each cohort were used as the dependent variable and neurotransmitter maps as the independent variables. This analysis involves fitting the same regression model on every possible combination of input variables. Dominance is computed as the average of the relative increase in the overall fit (adjusted $R^2$) when adding a single variable to the model. The collective dominance of all variables equals the total $R^2$. Next, dominance was normalized to the total $R^2$ to compare across models. Spin tests with 1000 permutations were used to assess statistical significance while preserving for spatial autocorrelation. Results were corrected using the FDR Benjamini-Hochberg procedure (Table S25).

## Autoradiography analysis

Autoradiography data was obtained from three post-mortem brains following the methodology outlined in ref. 77 and processed as described in reference[19]. In summary, densities of 15 different neurotransmitter receptors were determined in the supragranular, granular and infragranular layers in 44 cortical regions from three subjects ($n = 3$, two males, age range: 72–77 years) without neurological disease. The mean density of each receptor in each region was estimated[78] and a region-to-region mapping was performed manually with the 34 left hemispheric regions from the 68-region Desikan-Killiany atlas. Autoradiography neurotransmitter maps were used to predict connectivity alterations in HD cohorts through a dominance analysis as described in the previous section. Results were corrected using the FDR Benjamini-Hochberg procedure (Table S26). For more information about the methodology as well as the complete autoradiography dataset see ref. 19.

## Robustness analysis

To test the robustness of the results, MIND cortical similarity networks were obtained in the Track-HD cohort using the 318 region Desikan-Killiany and the Schaeffer 100, 200, and 400 parcellations. Cohen's $d$

for the differences in node strength between mHD and controls were estimated as previously described. There was a similar spatial pattern in node strength between the different parcellation resolutions (Fig. S2).

Since Desikan-Killiany 318-region is a subdivision of the Desikan-Killiany 68-region atlas, nodal strength Cohen's $d$ values from the finer parcellation were projected into the coarser parcellation and the results were correlated, finding a significant positive correlation ($rho = 0.47$, $P < 0.0001$).

Finally, ComBat, a batch-effect correction tool was applied to test whether harmonization would remove unwanted variation associated with site[41]. However, there were no significant site effects in nodal strength in any of the brain regions (Fig. S1). Therefore, we performed our analyses with the raw data.

### Reporting summary

Further information on research design is available in the Nature Portfolio Reporting Summary linked to this article.

## Data availability

The source data that supports the findings of this study are provided with this paper. Source data are provided with this paper. The TRACK-HD, Track-On and HD-YAS datasets used in this analysis have been deposited with the CHDI Foundation (https://chdifoundation.org/). Raw data are protected and are not available under privacy laws due to their sensitive and potentially identifiable nature. Biofluid samples will not be shared due to the limited amount of material available. Imaging and processed data are available following review by the CHDI Foundation subject to a proposal meeting the research criteria and demonstrating full GDPR compliance and agreement from the Principal Investigator, Professor Sarah Tabrizi. Brain surface images have been developed with the ENIGMA toolbox (https://enigma-toolbox.readthedocs.io/) Source data are provided with this paper.

## Code availability

All code used to perform these analyses is available at https://github.com/cestevezfraga/hd-mind-neurotransmitter.

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

## Acknowledgements

To the patients and their families. We would like to express our gratitude to Dr Raymund AC Roos for his significant contributions to the Track-HD and TrackOn-HD cohorts as a principal investigator. S.J.T. is partly supported by the UK Dementia Research Institute that receives its funding from DRI Ltd., funded by the UK Medical Research Council award number (DRI-TAP24/28). C.E.-F., S.G., R.I.S., G.R., and S.J.T. received support from a Wellcome Collaborative Award (200181/Z/15/Z). E.J.W. was funded by Medical Research Council (MR/M008592/1), European Huntington's Disease Network and CHDI Foundation. LMB was funded by a Medical Research Council Career Development Award (MR/W026686/1).

## Author contributions

C.E.-F., S.J.T., P.M., J.H., and I.S. conceived the present study. S.J.T., A.D., B.L., and B.R.L. designed the Track-HD and TrackOn-HD studies and supervised data collection. S.J.T. designed the HD-YAS study and supervised data collection. P.Z., R.I.S., S.G., E.B.J., E.J.W., and L.M.B. contributed to data collection in the HD-YAS cohort. E.J.W. and L.M.B. contributed to NfL data collection in the TRACK-HD cohort. C.E.-F., J.H., I.S., and P.M. performed the experiments. C.E.-F. and P. M. wrote the manuscript. C.E.-F., P.M., J.H., B.H., R.I.S., B.M., S.L.V., G.R., S.J.T., A.D., B.L., B.R.L., and S.J.T. edited the manuscript.

## Competing interests

The authors declare no competing interests.

## Additional information

[1]Department of Neurodegenerative Disease, University College London, London, UK. [2]Department of Psychiatry, University of Cambridge, Cambridge, UK. [3]Department of Computer Science and Technology, University of Cambridge, Cambridge, UK. [4]McConnell Brain Imaging Centre, Montréal Neurological Institute, McGill University, Montréal, QC, Canada. [5]Department of Psychiatry and Psychotherapy, University Hospital Tübingen, Tübingen, Germany. [6]German Centre for Mental Health, Tübingen, Germany. [7]Max Planck School of Cognition, Leipzig, Germany. [8]Southampton General Hospital, Southampton, UK. [9]Department for Science Innovation and Technology, London, UK. [10]Sorbonne Université, Paris Brain Institute (ICM), AP-HP, Inserm, CNRS, Pitié-Salpêtrière University Hospital, Paris, France. [11]University of Ulm, Ulm, Germany. [12]Department of Medical Genetics, Centre for Molecular Medicine and Therapeutics, University of British Columbia, Vancouver, BC, Canada. [13]UBC Children's Hospital, Vancouver, BC, Canada. [14]Institute of Neuroscience and Medicine, Research Centre Jülich, Jülich, Germany. [15]Institute of Systems Neuroscience, Medical Faculty, Heinrich Heine University Düsseldorf, Düsseldorf, Germany. [16]Otto Hahn Group "Cognitive Neurogenetics", Max Planck Institute for Human Cognitive and Brain Sciences, Leipzig, Germany. [17]Wellcome Centre for Human Neuroimaging, UCL Queen Square Institute of Neurology, University College London, London, UK. [18]Roche Pharma Research and Early Development (pRED), F. Hoffmann-La Roche Ltd., Basel, Switzerland. [19]These authors contributed equally: Sarah J. Tabrizi, Peter McColgan. ✉e-mail: estevezfragacarlos@gmail.com; s.tabrizi@ucl.ac.uk

