## [Transparent Peer Review file · Nature Communications]

Cell-specific mechanisms drive connectivity across the time course of Huntington's disease

Corresponding Author: Dr Carlos Estevez-Fraga

Version 0:

Reviewer comments:

Reviewer #1

(Remarks to the Author)

The application of a new connectivity system to participants with preHD and HD is noteworthy. Interpretations, however, of these data require further explanation for the readers. There are multiple "leaps" made from multiple data systems to the conclusions drawn. Perhaps the title could be more descriptive and less theoretical because the readers will not know the multiple derivations required to get to the conclusion. The greatest concern is the lack of citation for basic science findings supporting the conclusions made. There are multiple basic science studies of HD spreading and cell-autonomous and animal models testing these hypotheses. To make a basic science conclusion with imaging data might require a review of the basic science literature. Additional comments are provided here:

Overview

This is a timely neuroimaging paper from a highly experienced HD specialty group. Efforts to utilize the relatively newer Morphometric INverse Divergence (MIND) method rather than the prior approach of morphometric similarity networks (MSNs) is an excellent resource for the neuroimaging HD field. Given the novelty of the method, however, there are some components of the paper that might benefit from didactic writing and comparison between the two approaches regarding this specific medium spiny neuron focused neurodegenerative disease.

Methods

Details are critical for reproducibility and imaging tools are among those most difficult to replicate. To improve our ability to do this, journals are requesting more detail and information as shown in the guidelines for Nature Communication. Though it appears that the authors have provided most of the data required, this reader found it difficult to locate some basic information that would be beneficial to mention in the body of the paper and then refer the reader to the supplemental materials as desired. For example,

The data are from three studies and two of the studies are from four sites. What was the variation in site hardware, software, head coils, acquisition parameters, etc? How were potential site differences assessed and controlled? Were the controls all from the same studies?

Given that the paper uses normal control data from available datasets, it was often difficult to determine what was NC data acquired in the HD studies and what was data from the datasets.

What burden score was used and what is the rationale for the score used? Is the burden score published with cut-off values for early versus late preHD so all readers know what this means? If not, additional detail is required here.

If the definition for "epicentre" standard in the literature so its definition is determined by the data or findings from the data? I have the same question for all of the derived data used in the paper. Please cite what variables are established and validated in the literature and what values have been created for this paper. The definition of connectome and node varies and citations would provide assistance here for comparison of the derived variables with previous publications.

As far as I could find, no version information is provided for the usage of Freesurfer and recon-all. The Freesurfer version should be disclosed for the sake of replicability.

Statistical analyses

Given the high number of statistical comparisons made for this paper, it may be prudent to determine variations in p-values, such as the experiment-wise, hypothesis-wise, and analysis-wise correction rates. The authors mention usage of FDR for several of the analyses, but not all, and then report findings that failed FDR.

Nonsignificant findings should be deleted from the paper. The establishment of a p-value has been a standard in the field of biostatistics for a good reason. I find the mention of. Nonsignificant findings distracting and it makes me question other basic principles of statistics, such as normality of distributions and homogeneity of variances.

How are "spatial patterns" defined? Is there a standard in the field and if so, please cite the standard. Usage of standard atlases and data sharing is highly encouraged for imaging studies so we can better determine and harmonize location and other critical criteria to make attempt at "ground truth".

Functional Network Connectivity and HCP

The authors utilize the HCP data set to compute normative cortico-cortical and subcortico-cortical structural and functional connectivity matrices. The spatial correlation is then computed between the normative cortical and subcortical epicenter maps and the whole-brain d-maps for the MIND cortical similarity networks in the HD cohorts.

I have several questions about this:

No citation is given for the HCP data set, and as there are several releases of HCP such as HCP-Aging, HCP-Development etc., a specific citation needs to be provided.

The age range of HCP is quite narrow (22-36 years), while the age ranges in the HD cohorts are more broad (18-65 in Track-HD for example). How do the authors think this may confound the resulting correlation analysis given that the chosen HCP subset does not include any healthy aging adults? The authors may be interested in looking at the HCP-Aging cohort, which includes individuals with ages between 31 and 100.

No details about preprocessing or computation of the structural and functional connectivity measures from fMRI or diffusion data in HCP were included. Presumably the authors are using publicly released derivatives and not reprocessing the data. Even so, the authors should disclose as such and may benefit from including the preprocessing and analysis details, or at least providing a citation which outlines those steps.

It is a bit strange that the authors utilize structural and functional measures from a normative data set, and don't include any parallel analysis of MIND connectivity within the normative data set as well. Although MIND is computed for controls from the cohorts, it may be beneficial to perform the MIND connectivity analysis for HCP as well (just using Freesurfer and recon-all) even just as a sanity check, if not as a further normative analysis which would allow for additional comparisons. If there are any confounds in HCP which make this difficult, that should be disclosed as these would also presumably affect the comparison with the structural and functional connectivity measures from HCP.

The authors disclose in their discussion that they did not use any functional or diffusion imaging from the HD cohort because of limited sample sizes and heterogeneous scanning protocols. I take this to mean that the three studies included here did not obtain any functional or diffusion imaging. While the argument for not including heterogeneous studies makes sense, more details should be provided to bolster this argument in order to justify why COMBAT and other harmonization tools would not work for getting around these issues. This work would certainly be bolstered by including additional functional and diffusion analysis in HD, and it seems the authors do not pay much mind to this rather significant exclusion of modalities (especially given the use of functional and diffusion derivatives from normative studies!).

(Remarks on code availability)

Reviewer #2

(Remarks to the Author)

(Remarks on code availability)

There is a README file available; however, it is quite sparse. All of the python code is provided in notebooks with hard-coded paths to data, and no information is provided about filepath organization, CSV file format, or any other requirements that are needed to run this code. Furthermore, no documentation is provided in the python notebooks explaining what each block of code does. The R and MATLAB scripts have some better documentation; however, also hard-code filepaths without any documentation describing how data should be stored in those paths. Inline documentation is provided in the R and MATLAB files, which helps some.

Without example data or much of an idea how data was to be organized, I was not able to run the code or reproduce any

results. While data release might not be possible, I would encourage the authors to include an example data set from an open-source location such as OpenNeuro.org, and to provide instructions with how to run each part of the code individually. Better yet, the authors could provide some information about how to produce specific analyses performed in their work.

Reviewer #3

(Remarks to the Author)

In this manuscript, the authors used Morphometric INverse Divergence (MIND) to study the degree of divergence in connectivity patterns in controls versus Huntington's Disease (HD) patients along the disease progression time course including early premanifest, late premanifest, and manifest HD. Utilizing T1-weighted MRI data from three HD study cohorts HD-YAS (57 early preHD, 60 controls), TrackOn-HD (85 late preHD, 89 controls), and Track-HD (110 manifest HD and 111 controls), the authors generated MIND networks. Group-level analysis results of these data revealed disease epicenters and correlations with plasma NFL measurements, and neurotransmitter distributions were performed to study organizational principles, receptor gradients, and neurotransmitter systems.

This is an interesting paper suitable for publication in Nature Communications. The data presented in this paper support the main conclusion and reveal interesting details of connectivity alterations in HD.

The manuscript would be improved by considering the following points:

Page 6

Lines 11 -16

"The Morphometric INverse Divergence (MIND) technique is a new method to investigate brain connectivity that leverages multiple imaging features obtained from T1-weighted MRI such as cortical thickness, surface area, mean curvature, sulcal depth and gray matter volume. MIND-derived brain networks are closely associated with biological characteristics such as gene expression, being a useful method to investigate the pathophysiological mechanisms involved in connectivity alterations in HD17."

It is unclear from this paragraph, yet it is an important point for this paper, why MIND-derived networks are closely associated with gene expression. The reader would benefit from more information and details on the original MIND reference #17, in which the term was coined.

Lines 20-21

"We hypothesized the existence of hyperconnectivity in HD participants very far from disease onset, progressing to hypoconnectivity through later stages."

Specify "very far" or refrain from making an emphasized yet vague statement.

Page 9

Lines 13-14

"Robustness analyses were performed to examine the impact of ComBat harmonization as well as different parcellation resolutions (Figures S1-S2)"

Consider explaining ComBat harmonization as equivalent to batch effect correction, removing systematic differences that are caused by different sites or scanners for example, while leaving biological signal intact.

Page 10

Lines 1-3

"For connection strength (Figure 2, Tables S2-S5) in early preHD, 21.67 years from disease onset, NBS analyses revealed significant increases in connectivity relative to healthy controls (PFWE=0.029); no decreases in connectivity were observed."

The authors list all significant and non-significant p-values throughout their manuscript. Consider adding this non-significant p-value to the text as well.

Lines 6-7

"Increased connectivity was also seen in five connections in mHD (PFWE=0.022), localized to the occipital cortex (Figure S3)."

Figure S3 shows the hyperconnectivity of four occipital cortical regions and one parietal (superior parietal) region. Please correct the statement accordingly.

Page 11

Lines 5-8

"These findings strongly suggest that the areas with increased connectivity during very early stages of the disease are the

same brain regions that then experience connectivity loss later in the disease course.”

If these findings strongly suggest that the same areas that show increased connectivity in early preHD are the same that experience connectivity loss later in the disease - then couldn't / shouldn't there be an observable and significant inverse correlation between node strength in early preHD and mHD?

Page 14

Lines 4-6 and lines 27-29

“In addition, the structural and functional connectivity patterns of specific regions (“epicentres”) has been proposed to align with disease-associated atrophy, indicating that transneuronal spread of pathogenic factors contributes to neural death in neurodegenerative diseases such as frontotemporal dementia or Alzheimer’s disease 28.”

“These results suggest that the relative contribution of axonal connections varies during the course of the disease, being particularly prominent during late preHD stages, when potentially pathogenic factors such as the misfolded mHTT could spread along axons in highly connected regions²⁹. This phenomenon is region-specific, targeting paracentral and cingulate cortices, which are known to develop early cortical gray matter loss in HD²³.”

Also page 17

Lines 18-20

“Shortly before motor onset, axonal spread of pathogenic factors could contribute to accelerated atrophy.”

As there is no direct and definite proof for the spread of pathogenic factors along axons, alternative hypotheses should also be discussed. For example, synaptic dysfunction, leading to aberrant neuronal signaling, can result in degeneration.

Page 19

Lines 6-8

“PET neurotransmitter data for 19 receptors and transporters from more than 1,200 healthy controls was parcellated using the Desikan-Killiany atlas²⁶ obtaining a 68-brain region by 19-neurotransmitter matrix¹⁵.”

Please list here the neurotransmitter families comprising these 19 neurotransmitters (Serotonin, Dopamine, GABA, etc.)

Page 25

Lines 20-23

“Our previous work has shown that changes in the occipital cortex of pwHD are largely developmental rather than neurodegenerative³³ and occipital hyperconnectivity could be related to different mechanisms governing regions densely connected to the striatum, or compensation.”

Consider rewording for clarity. Suggestion:

“Our previous work has shown that changes in the occipital cortex of pwHD are largely developmental rather than neurodegenerative³³ and occipital hyperconnectivity could be related to compensation, or different mechanisms governing regions densely connected to the striatum.”

Page 26

Lines 25-27

“Interestingly, a recent study has shown that medium spiny neurons, but not other striatal cells, undergo somatic expansion, which subsequently triggers severe changes in gene expression in these neurons leading to neuronal death and atrophy³⁹.”

A second study, already mentioned in the manuscript as reference number 51 (Mätlik K, Baffuto M, Kus L, et al. Cell-type-specific CAG repeat expansions and toxicity of mutant Huntingtin in human striatum and cerebellum. *Nat Genet* 2024; 56: 383– 94.) , should be added to this statement alongside reference number 39 (Handsaker RE, Kashin S, Reed NM, et al. Long somatic DNA-repeat expansion drives neurodegeneration in Huntington disease. *bioRxiv* 2024; : 2024.05.17.592722.).

Figure 1

Please organize sections in marked figure panels (A-Z) and match with text for improved interpretability and readability.

Figure 2

Resolution low, small font hardly readable, i.e. in panel a (group network connection strength values) and e (cortical region annotations).

Redundant word “were” in the figure text:

d, Violin plots depicting ROIs showing significant differences at the uncorrected level in early preHD and late preHD. In mHD, for visualization purposes, only the top five ROIs were with larger effect sizes were shown, all of them significant

following FDR correction at a $P < 0.05$.

Figure 3

“b, Scatterplots representing the correlations between plasma NfL and nodal strength for MIND connectivity in early preHD, late preHD and mHD. The ROI with largest absolute magnitude in the correlation coefficient was selected for each cohort for visualisation purposes.”

The selection of the ROIs with the largest absolute magnitude for visualization purposes is unclear. Appreciating the presentation of the distribution of the individual data points that form the largest correlation coefficients and underlie the brain heat map extremes in panel a, I suggest elaborating on and explaining better the intention of the “visualization purposes” in the figure text.

General

Please add to the discussion a comment on how a left-right distribution of significant associations between connectivity and plasma NfL as shown in Figure 3, panel c can be explained.

Figure 4

General

Given the pronounced vulnerability and degeneration observed in the striatum in Huntington’s Disease, the absence of distinct subcortical epicenters is unexpected. This observation raises important questions about underlying disease mechanisms or potential methodological factors, warranting further discussion by the authors.

(Remarks on code availability)

Reviewer #4

(Remarks to the Author)

(Remarks on code availability)

Version 1:

Reviewer comments:

Reviewer #1

(Remarks to the Author)

The revised manuscript is excellent. The authors responded to our previous review's input and have strengthened the paper. These findings are presented more clearly and sufficiently cited. Any remaining limitations in this research will be addressed by what will hopefully be a strategic response from the field to embrace this paper and follow on to advance understanding of HD further using connectivity and neuroimaging. Congratulations.

(Remarks on code availability)

Not applicable.

Reviewer #2

(Remarks to the Author)

(Remarks on code availability)

Reviewer #3

(Remarks to the Author)

I have carefully reviewed the revisions made by the authors in response to the comments provided during the initial review.

The authors have satisfactorily addressed the concerns raised and made significant improvements to the manuscript.

I believe this revised version meets the standards for publication and represents an important contribution to the field.

However, I recommend correcting the formatting of Table S22 P values in the epicenter analysis, before the manuscript is published.

(Remarks on code availability)

Reviewer #4

(Remarks to the Author)

(Remarks on code availability)

Please note that we are using the line numbers from the PDF generated after uploading the manuscript to the submission platform.

REVIEWERS' COMMENTS

REVIEWER 1

REMARKS TO THE AUTHOR

The application of a new connectivity system to participants with preHD and HD is noteworthy. Interpretations, however, of these data require further explanation for the readers. There are multiple "leaps" made from multiple data systems to the conclusions drawn. Perhaps the title could be more descriptive and less theoretical because the readers will not know the multiple derivations required to get to the conclusion. The greatest concern is the lack of citation for basic science findings supporting the conclusions made. There are multiple basic science studies of HD spreading and cell-autonomous and animal models testing these hypotheses. To make a basic science conclusion with imaging data might require a review of the basic science literature.

RESPONSE TO REMARKS TO THE AUTHOR

Many thanks for these comments. We have now edited the title as follows: "Cell-specific mechanisms drive connectivity across the time course of Huntington's disease". We feel this is more descriptive and less theoretical than the previous version and hope this addresses the reviewers concerns.

We agree with the reviewer that many of the inferences made in our manuscript are derived from basic science studies that deserve citation. We have now added multiple new basic science citations throughout the paper wherever relevant concepts are discussed. Please note that we had to remove other redundant references to comply with the Journal's policy, as there is a maximum of 70 references:

Vonsattel 1985 (Reference #2)
Hawrylycz 2012 (Reference #10)
Hawrylycz 2015 (Reference #11)
Raj 2012 (Reference #12)
Tremblay 2021 (Reference #13)
Zheng 2019 (Reference #14)
Van Roon-Mom 2002 (Reference #24)
Creus-Muncunill 2019 (Reference #21)
Stöberl 2023 (Reference #22)
Thomas 2011 (Reference #23)
Oliveira 2010 (Reference #25)
Altar 1997 (Reference #26)
Zuccato 2005 (Reference #27)
Hsiao 2013 (Reference #28)
Crotti 2014 (Reference #29)
Rubinov & Sporns (Reference #44)
Mouro Pinto 2020 (Reference #55)
Neueder 2014 (Reference #68)

Rockland 1979 (Reference #62)

Finally, as suggested, we have also expanded our explanation about cell-autonomous mechanisms versus non-cell-autonomous mechanisms in Huntington's disease extensively referencing basic science research:

"(...) pointing towards cell-autonomous mechanisms (i.e. processes within a cell regulated by its own genes and biological features, independently of other cells) being associated with neuronal death.

Cell-autonomous mechanisms influence the pattern of neuronal loss in HD. Moreover, the mHTT protein sequesters transcription factors resulting in transcriptional dysregulation, another core pathogenic mechanism in HD. Similarly, mHTT alters mitochondrial dynamics interfering with ATP production and calcium buffering.

There is also extensive evidence of non-cell-autonomous mechanisms such as abnormal connectivity resulting in neuronal degeneration. Brain Derived Neurotrophic Factor (BDNF) is synthesized in cortical neurons but not in medium spiny neurons, and needs to be transported to the striatum. Cortico-striatal degeneration disrupts this process facilitating striatal atrophy. Also, alterations in non-neuronal cells can result in neuronal dysfunction and death. Expression of mHTT in astrocytes leads to motor deficits, and astrocyte pathology can exacerbate neuronal dysfunction through alterations in glutamatergic mechanisms. Similarly, the expression of mHTT in microglia induces neuronal death through non-cell-autonomous mechanisms.

However, the relative contribution of cell-intrinsic mechanisms and non-cell-autonomous mechanisms is possibly specific to disease and stage, and both approaches have not been evaluated thoroughly -or simultaneously- in HD human brains." (from page 6 line 113 to page 7 line 135)

COMMENT 1

Additional comments are provided here:

Overview

This is a timely neuroimaging paper from a highly experienced HD specialty group. Efforts to utilize the relatively newer Morphometric INverse Divergence (MIND) method rather than the prior approach of morphometric similarity networks (MSNs) is an excellent resource for the neuroimaging HD field. Given the novelty of the method, however, there are some components of the paper that might benefit from didactic writing and comparison between the two approaches regarding this specific medium spiny neuron focused neurodegenerative disease.

RESPONSE TO COMMENT 1:

Many thanks for these comments. We agree that the reader might benefit from additional clarifications about these methods. We have therefore added an explanation about morphometric similarity networks (MSNs) as well as discussed the differences between morphometric similarity mapping and MIND.

Importantly, although striatal atrophy caused by degeneration of medium spiny neurons is the pathological hallmark of HD, there is extensive evidence showing widespread cortical involvement in the disease, starting before the emergence of motor symptoms. This is consistent with our results demonstrating gradual alterations in cortico-cortical connectivity across the timecourse of the disease. We have also clarified this important aspect.

“Traditionally, brain structure has been investigated through univariate analyses of isolated parameters such as fractional anisotropy or sulcal depth in specific areas. However, brain regions do not function in isolation but form complex networks both at the cellular level and systems level. Understanding large-scale network architecture is therefore essential. Morphometric similarity mapping is a method to develop structural morphometric similarity networks (MSNs) using information from different micro- and macrostructural parameters through representing each brain region as a vector, followed by the pairwise correlations across brain regions. Morphometric similarity networks are associated with gene expression, recapitulate cortical cytoarchitectonic areas and are also related to axonal connectivity. Alterations in MSNs are present in schizophrenic individuals, as well as in people with Alzheimer’s disease. However, MSNs depend on summary statistics for each brain region instead of vertex-level data, and use Z-scores, constraining variability across brain regions.

The Morphometric INverse Divergence (MIND) technique is a new method to investigate global brain connectivity that also leverages multiple imaging parameters at the vertex-wise level. In MIND, the similarity between brain regions is estimated through the Kullback-Leibler divergence between their distributions. The MIND method has been successfully applied to different cohorts being more consistent with principles of cortical organization than other methods such as morphometric similarity networks (MSNs). In healthy brains, highly connected regions have a similar transcriptional profile. A gene co-expression network derived from the Allen Human Brain Atlas (AHBA) showed a stronger correlation with MIND-derived connectomes compared to diffusion tensor imaging or MSNs. In addition, MIND-derived connectivity metrics were more closely associated with twin-based and SNP-based heritabilities than other imaging phenotypes, particularly in primary sensory regions. Brain networks estimated through MIND are correlated with axonal connectivity and cortical cytoarchitecture, being also more sensitive to predict age than similar techniques through machine learning models. Importantly, MIND can be used with imaging features obtained from simple T1-weighted MRI sequences such as cortical thickness, surface area, mean curvature, sulcal depth and gray matter volume. These characteristics support the use of MIND to investigate connectivity across the time course of in HD.” (From page 7 line 137 to page 8 line 167)

“The disease is caused by an expansion of the cytosine-adenine-guanine (CAG) trinucleotide repeat in the first exon of the HTT gene, encoding for the mutant Huntingtin protein (mHTT) resulting in marked neuronal loss affecting medium spiny neurons in the striatum. Widespread cortical atrophy also occurs, starting long before the onset of motor symptoms” (Page 5 lines 79-83)

COMMENT 2:

Methods

Details are critical for reproducibility and imaging tools are among those most difficult to replicate. To improve our ability to do this, journals are requesting more detail and information as shown in the guidelines for Nature Communication. Though it appears that the authors have provided most of the data required, this reader found it difficult to locate some basic information that would be beneficial to mention in the body of the paper and then refer the reader to the supplemental materials as desired. For example,

The data are from three studies and two of the studies are from four sites. What was the variation in site hardware, software, head coils, acquisition parameters, etc? How were potential site differences assessed and controlled? Were the controls all from the same studies?

RESPONSE TO COMMENT 2:

We have now added the additional information requested by the reviewer in the main manuscript, under a new subsection in the Results entitled “Study population and MRI data”. Each of the three cohorts used in this paper (TrackHD, TrackOn-HD and HD-YAS) included persons with Huntington’s disease (pwHD) and healthy controls. In this revision we clarify that, to determine Cohen’s d for MIND connectivity, data from controls from each cohort was obtained. These controls were age- sex-matched with pwHD from the same study. In addition we now expand on the number of sites, scans and sequences used in these studies. We also detail how we have assessed and controlled site differences and their impact by using combat harmonisation and including site as a covariate in our analyses, respectively.

We also include additional information in specific Methods subsections.

See below:

“Study population and MRI data

512 participants were included, comprising 252 pwHD and 260 controls. These data were obtained from three different studies: HD-YAS (57 early preHD and 60 controls), TrackOn-HD (85 late preHD and 89 controls) and Track HD (110 mHD and 111 controls) (Figure 1A). Controls were age- and sex- matched with pwHD within each cohort. PreHD individuals were required to have a Total motor score (TMS) < 5 and mHD individuals were required to score ≥ 5 in the same scale. See Table S1 for participant demographics.

HD-YAS was a single centre study carried out in the London site in a Siemens scan. In contrast, TrackHD and Track-On HD included participants from four study sites (Leiden, London, Paris and Vancouver). In Track-HD and TrackOn-HD 3T-MRI sequences were acquired in two scanner systems: Philips in Leiden and Vancouver and Siemens in London and Paris. T1-weighted data were acquired using a 3D Magnetization prepared rapid gradient echo (MPRAGE) sequence in all studies.

ComBat harmonization, an empirical Bayesian method for data harmonization to evaluate non-biological variance caused by differences in MRI scanners and acquisition parameters did not reveal significant differences in any brain region

(Figure S1, Table SX). Therefore raw data was used in this study. Site was added as a covariate in all analyses.

For further details please see the Methods section (MRI data acquisition and MRI data processing subsections)”
(Page 11 lines 211-233)

COMMENT 3:

Given that the paper uses normal control data from available datasets, it was often difficult to determine what was NC data acquired in the HD studies and what was data from the datasets.

RESPONSE TO COMMENT 3:

We agree with Reviewer #1 that in the initial version of this paper it was difficult to determine the origin of the data while reading the Results section, which was only fully detailed in the Methods section.

In this research, we have used data available in-house from both healthy controls and pwHD in the TrackHD, TrackOn-HD and HD-YAS cohorts alongside information from healthy controls obtained from publicly available data sources (eg PET data from Hansen JY et al. Nature Neuroscience 2022, PMID: 36303070, or connectivity data from ENIGMA for the epicentre analysis: <https://enigma.ini.usc.edu/about-2/>)

Initially, we computed Cohen’s *d* data using data from pwHD and healthy controls from the TrackHD, TrackOn-HD and HD-YAS cohorts. These results were then related to normative data from external sources in subsequent analyses.

In this new version of the manuscript we clarify these aspects in the Results section, before each individual analysis:

*“For node strength and connection strength, HD-specific MIND connectivity alterations were estimated through adjusted Cohen’s *d*, providing a numeric value for the node strength and connection strength difference between pwHD and controls from each cohort (HD-YAS -early preHD-, TrackOn-HD -late preHD- and TrackHD -mHD-) in each node and in each connection.”*(Page 12, lines 253-256)

“producing symmetrical connectivity matrices for each participant in TrackHD, TrackOn-HD and HD-YAS” (Page 12, lines 243-244)

“To investigate the presence of disease epicentres in HD, HD-specific nodal strength data from the previous section was correlated with structural and functional connectivity matrices from healthy controls generated from the Human Connectome Project dataset.” (Page 17, lines 329-332)

“We explored associations with different organizational principles, obtained from datasets of healthy controls different from the HD-YAS, TrackOn-HD and Track-HD cohorts.” (Page 20, lines 377-378)

“PET neurotransmitter data for 19 receptors and transporters from more than 1,200 healthy controls was parcellated using the Desikan-Killiany atlas obtaining a 68-brain region by 19-neurotransmitter matrix and compared with HD-specific nodal strength.” (Page 23, lines 436-439)

“A total of 19 neurotransmitter PET maps from healthy controls were investigated and the association with HD-specific nodal strength was computed through the $R^2_{adjusted}$ of each multilinear regression model was tested using FDR-corrected spin tests.” (Page 26, lines 481-484)

We have also tried expand the rationale for our approach in the introduction:

“Organizational principles in the healthy brain can be used to understand multiscale systems-level disease mechanisms in neurodegeneration where regions affected at later stages during the course of the disease are connected with areas that degenerated first. Healthy white matter networks in controls without brain disease overlap with atrophy patterns across psychiatric and neurodegenerative diseases such as schizophrenia and Alzheimer’s disease, with lesions concentrated in hub regions. Similarly, disease-specific epicentres estimated from task-free functional MRI data provide systems-level evidence for transneuronal spread of pathogenic proteins frontotemporal dementia and cortico-basal syndrome. In pwHD, a network diffusion model suggests that proximity is a key factor in the spread of pathology across adjacent neurons.” (Page 5 lines 92-101)

COMMENT 4

What burden score was used and what is the rationale for the score used? Is the burden score published with cut-off values for early versus late preHD so all readers know what this means? If not, additional detail is required here.

RESPONSE TO COMMENT 4:

Unfortunately, there are no pre-established cut-off thresholds to define categories based on clinical scales. In TrackHD and TrackOn-HD manifest and premanifest individuals were defined based on the total motor score (TMS). This is the main scale used to evaluate motor function in HD, scoring from 0 (normal) to 124 (most severe symptoms). The TMS had to be > 5 in manifest HD (mHD) and ≤ 5 preHD participants. The disease burden score (DBS) is a function of age and CAG repeat ($DBS = Age * (CAG - 35.5)$) that quantifies the total pathogenic burden of HD. Both mHD and preHD participants from TrackHD and TrackOn-HD were required to have a $DBS \geq 250$.

In contrast, preHD participants from HD-YAS were also required to have a $TMS < 5$ as well as a $DBS \leq 240$.

In clinical practice, people with HD (pwHD) are classified as having premanifest HD or manifest HD through the diagnostic confidence level (DCL) scale. The DCL ranges from normal ($DCL=0$) to non-specific motor abnormalities ($DCL=1$), followed by motor abnormalities that may be signs of HD with 50-89% confidence ($DCL=2$), motor signs that are likely signs of HD with 90-98% confidence ($DCL=3$) and motor impairments that are unequivocal signs of HD with 99% confidence ($DCL=4$).

Patients with DCL 0-1 are defined as preHD, DCL = 2-3 as prodromal HD, and DCL = 4 as manifest HD (Huntington's Study Group, 1996). However, there is a substantial subjective component as DCL scores do not map onto DBS scores which is a prognostic, rather than a diagnostic scale.

We now clarify the TMS threshold in the Results section:

“PreHD individuals were required to have a Total Motor Score (TMS) ≤ 5 and mHD individuals were required to score > 5 in the same scale.” (Page 11, lines 215-217)

COMMENT 5:

If the definition for “epicentre” standard in the literature so its definition is determined by the data or findings from the data? I have the same question for all of the derived data used in the paper. Please cite what variables are established and validated in the literature and what values have been created for this paper.

RESPONSE TO COMMENT 5:

The analysis performed in this section was originally developed in the paper by Larivière's et al (Larivière's et al. Science Translational Medicine, 2020 PMID: 33208365). There, the term “epicenter” is aligned with the original definition from Zhou et al. (Zhou et al. Neuron 2012, PMID: 22445348) where epicentres are defined as “regions whose connectivity patterns—in the healthy brain—most closely mirrored the disease vulnerability pattern”.

From an operational perspective, in their paper, Larivière et al defined as epicenters regions where there is a significant overlap between the connectivity pattern in healthy controls and in people with epilepsy.

We have now added this definition in the current version of the manuscript.

“Disease “epicentres” are brain areas where there is a significant overlap between the connectivity pattern in healthy controls and disease-specific connectivity alterations.” (Page 17, lines 321-322)

COMMENT 6

The definition of connectome and node varies and citations would provide assistance here for comparison of the derived variables with previous publications.

RESPONSE TO COMMENT 6:

We have used the definition of connectome developed by Rubinov (Rubinov & Sporns Neuroimage 2010, PMID: 19819337) and Sporns et al (Sporns et al. PLoS Computational Biology 2005, PMID: 16201007). We defined a brain region as a ‘node’. A ‘connectome’ is formed by brain regions (nodes) and connections.

We have now included the definition described above, together with the mentioned citations:

“In brain connectivity, the term ‘node’ refers to a brain regions. The brain ‘connectome’ is composed of all brain regions (nodes) and the connections between them (Page 12, lines 237-238)

COMMENT 7:

As far as I could find, no version information is provided for the usage of Freesurfer and recon-all. The Freesurfer version should be disclosed for the sake of replicability.

RESPONSE TO COMMENT 7:

We have used FreeSurfer version 7. We have now made this clear every time the FreeSurfer software is mentioned during in the manuscript. In this revision we also now provide a reference for FreeSurfer software (Fischl et al. Neuroimage 2013, PMID: 22248573)

“T1-weighted MRI-derived structural features parcellated through FreeSurfer’s version 7 recon-all command were processed using the MIND approach to generate a connection measure of inter-regional similarity between two cortical areas across brain regions,” (Page 12, lines 240-241)

“Baseline 3D T1 volumetric images were bias-corrected using the N3 algorithm. Next, FreeSurfer version 7” (Page 37, lines 774-775)

“We used the mesh reconstructions of the cortical surface generated from T1-weighted MRI scans using FreeSurfer’s version 7 recon-all command.” (Page 38, lines 788-789)

“Structural T1-weighted brain MRI scans were parcellated using the 68 cortical region Desikan-Killiany atlas with FreeSurfer’s version 7 recon-all command.” (Page 10 lines 199-200)

COMMENT 8:

Statistical analyses

Given the high number of statistical comparisons made for this paper, it may be prudent to determine variations in p-values, such as the experiment-wise, hypothesis-wise, and analysis-wise correction rates. The authors mention usage of FDR for several of the analyses, but not all, and then report findings that failed FDR.

RESPONSE TO COMMENT 8:

Many thanks for this. We have now gone through the entirety of the P values in the paper and added these in the supplementary. We have now also generated not only the FDR-corrected P_{spin} values, but also the uncorrected P values. Given the large number of tests performed, a single table including all the experiments would be extremely long (around 30 pages). Therefore, we have included several additional new tables to the supplementary including:

- A table with the raw P/FDR-corrected values before / after ComBat harmonization (Table S2)

- A table with the raw/FDR-corrected P_{spin} values from the epicentre analysis. (Table S22)
- A table with the raw/FDR-corrected P_{spin} values from the organizational principles analysis (Table S23)
- A table with the raw/FDR-corrected P_{spin} values from the PET neurotransmitter analysis (Table S25)
- A table with the raw/FDR-corrected P_{spin} values from the autoradiography neurotransmitter analysis (Table S26)

Network-based statistics provides a single P value after family-wise error rate correction. We now provide these P values in the title from tables S4-S7.

We have also noticed that the organizational principles and receptome analysis had not been FDR-corrected in the previous version. This was caused as the original scripts did not perform correction for multiple comparisons. However, the correction had only a minimal impact in the results and does not affect the conclusions of the paper. After, FDR-correction receptome gradient 1 was still significantly correlated with nodal strength in early preHD and late preHD, while, as in the previous version of this manuscript, there were no additional significant correlations in the receptome analysis.

The same organizational principles except MEG in late preHD are significant. We have amended figure 5 accordingly.

Please find remaining P values in tables S20 to S26.

COMMENT 9:

Nonsignificant findings should be deleted from the paper. The establishment of a p-value has been a standard in the field of biostatistics for a good reason. I find the mention of. Nonsignificant findings distracting and it makes me question other basic principles of statistics, such as normality of distributions and homogeneity of variances.

RESPONSE TO COMMENT 9:

We have now edited the text to remove all non-significant P values from the paper.

“For late preHD we did not find significant increases or decreases in connectivity.”
(Page 13, lines 269-270)

“Results at the node level were consistent with those observed at the connection level. In early preHD and in late preHD, there were no significant increases or decreases in node strength relative to healthy controls after FDR correction. For mHD, significant FDR-corrected reductions in connectivity compared to healthy controls were observed in 48 out of 68 brain regions ($P_{\text{FDR}} < 0.05$). No increases in connectivity were observed (Tables S8-S10). There were no correlations between NfL and node strength in early or late preHD after FDR correction. For mHD, significant FDR-corrected negative correlations with NfL were observed in 24 out of 68 brain regions ($P_{\text{FDR}} < 0.05$) (Table S11).” (Page 13, lines 283-290)

“ (...) the spatial pattern of node strength was significantly correlated between early preHD and late preHD ($\rho=0.26$, $P_{FDR}=0.046$). There was no correlation between node strength in early preHD and mHD but the spatial pattern of node strength was also significantly correlated between late preHD and mHD ($\rho=0.38$, $P_{FDR}=0.039$).” (from page 13, line 295 to page 14 line 298)

“ In early preHD, significant associations were seen for neurotransmitter systems ($\rho=0.44$, $P_{spin}=0.047$) but not gene expression. In late preHD, significant associations were seen for gene expression ($\rho=0.58$, $P_{spin}=0.031$) but not neurotransmitter systems. Similarly, in mHD a significant association was seen for gene expression ($\rho=0.42$, $P_{spin}=0.031$) but not neurotransmitter systems” (Page 21, lines 404 - 409)

COMMENT 10:

How are “spatial patterns” defined? Is there a standard in the field and if so, please cite the standard. Usage of standard atlases and data sharing is highly encouraged for imaging studies so we can better determine and harmonize location and other critical criteria to make attempt at “ground truth”.

RESPONSE TO COMMENT 10:

The term ‘spatial pattern’ was used to describe the distribution of MIND connectivity changes across the brain. We have now described this explicitly in the results section:

“The regional distribution of differences (i.e. spatial pattern) in MIND node strength” (Page 13 line 292)

Here, we have used the Desikan-Killiany, 68-cortical region parcellation atlas. The Desikan-Killiany atlas is widely used across literature, being the standard output of Freesurfer’s *recon-all* command. This is the parcellation applied in the original paper where MIND was described. Similarly, this is the parcellation available from the ENIGMA dataset, where the epicentre analysis was initially developed. In addition, the PET and autoradiography data used in this paper were similarly parcellated using the Desikan-Killiany 68-region cortical atlas in the original research.

To ensure that our results were not dependent on parcellation, we performed a robustness analysis confirming our findings in the more granular Schaeffer 100, 200 and 400 parcellations (Figure S2). These more granular parcellations were used for visual inspection of the spatial pattern of MIND connectivity in the mHD cohort. We have now cited the original reference of the Schaefer atlas.

In line with the changes suggested in previous sections by this Reviewer we have also cited the atlas reference each time every time it is mentioned (Desikan et al 2005, PMID: 16530430)

COMMENT 11:

Functional Network Connectivity and HCP

The authors utilize the HCP data set to compute normative cortico-cortical and

subcortico-cortical structural and functional connectivity matrices. The spatial correlation is then computed between the normative cortical and subcortical epicenter maps and the whole-brain d-maps for the MIND cortical similarity networks in the HD cohorts.

I have several questions about this:

No citation is given for the HCP data set, and as there are several releases of HCP such as HCP-Aging, HCP-Development etc., a specific citation needs to be provided. The age range of HCP is quite narrow (22-36 years), while the age ranges in the HD cohorts are more broad (18-65 in Track-HD for example). How do the authors think this may confound the resulting correlation analysis given that the chosen HCP subset does not include any healthy aging adults? The authors may be interested in looking at the HCP-Aging cohort, which includes individuals with ages between 31 and 100.

RESPONSE TO COMMENT 11:

We have now included a citation for the HCP datasets. See Page XX Lines XX

In this study we leverage multiple datasets derived from the healthy human brain to understand how the regional distribution of connectivity changes in HD relates to the organisational principles in the healthy brain. This is an approach which was originally pioneered by William Seeley (Seeley WW et al. Neuron, 2009 PMID: 19376066; Zhou, (...) Seeley Neuron, 2012 PMID: 22445348) and has been used extensively by many groups since then. We appreciate that age would be a confounder if conducting a group-wise analysis comparing HD individuals to those from the HCP. However the objective in our approach is to compare the regional distribution of cortical pathology in HD to cortical organisational principles in the healthy brain. Thus, using an older age cohort, such as the HCP-Aging cohort, would introduce the possibility of age-related pathology, that may perturb the organisational principles of the healthy brain.

COMMENT 12:

No details about preprocessing or computation of the structural and functional connectivity measures from fMRI or diffusion data in HCP were included. Presumably the authors are using publicly released derivatives and not reprocessing the data. Even so, the authors should disclose as such and may benefit from including the preprocessing and analysis details, or at least providing a citation which outlines those steps.

RESPONSE TO COMMENT 12:

In the first version of our manuscript we tried to clarify these details in the Methods section, referring the reader to the original papers. In this new version of the manuscript we have cited the original papers as well as provided additional details about the fMRI and diffusion data in the Results section.

In the epicentre analysis we have used data minimally pre-processed as described in Glasser et al. (2013), which we have now also cited. We have also included more details in the Results section and referred the reader to the Methods for more information:

“HCP resting-state fMRI and structural data underwent minimal preprocessing as described in Glasser et al. and was parcellated using the Desikan-Killiany 68-region cortical atlas. See the Methods section, subsection “Epicentre analysis” for additional information.” (Page 17 lines 334-339)

Similarly, we have expanded the Methods section to clarify how the pre-processing was performed instead of focusing exclusively on generation of the normative networks -which we have also expanded on significantly on. We have also added citations for the different tools used during the pre-processing and processing steps including MRtrix3, SIFT2 or FMRIB’s ICA-based X-noiseifier:

“Data already processed from the HCP dataset, including a group of healthy adults (n = 207, 83 males, age range 22 to 36 years) was used to generate normative cortico-cortical and subcortico-cortical structural and functional connectivity matrices as described in Larivière et al. Diffusion MRI data from the HCP dataset was intensity-normalized with the mean B0 image. Next diffusion data was corrected for distortions caused by motion and eddy currents as well as for susceptibility artifacts.

Resting-state fMRI data from the HCP dataset underwent first distortion as well as subject-motion correction. Next, resting-state fMRI data was corrected for magnetic field bias, skull removed and intensity-normalized before being mapped to the MNI152 space. FMRIB’s ICA-based X-noiseifier (FIX) was used to automatically remove additional noise components related to presence of white matter, pulsation or movement. For additional information about preprocessing see Glasser et al. Finally, diffusion MRI and resting-state fMRI data from the HCP dataset were parcellated with the 68-cortical and 18-subcortical regions of the Desikan-Killiany atlas.

Normative structural connectivity matrices were generated using MRtrix3. T1-weighted images were used to perform anatomically constrained tractography. Forty million streamlines were produced with a fractional anisotropy threshold of 0.06 and a maximum tract length of 250. The spherical-deconvolution informed filtering of tractogram (SIFT2) algorithm was then used to reconstruct whole-brain streamlines weighted by the cross-sectional multipliers. These were subsequently mapped onto the 68-cortical and 14 subcortical regions from the Desikan-Killiany atlas to generate subject specific structural connectivity matrices. A distance-dependent thresholding procedure was used to estimate group-averages followed by log transformation.

Normative functional connectivity matrices were generated computing the correlations between the time series of 68-cortical regions with all cortical and subcortical regions. Negative connections were zeroed. Group averaged functional connectomes were computed through z-transformation of aggregated subject-specific connectivity matrices.”(from page 39 line 838 to page 40 line 867)

COMMENT 13:

It is a bit strange that the authors utilize structural and functional measures from a normative data set, and don't include any parallel analysis of MIND connectivity within the normative data set as well. Although MIND is computed for controls from the cohorts, it may be beneficial to perform the MIND connectivity analysis for HCP as well (just using Freesurfer and recon-all) even just as a sanity check, if not as a further normative analysis which would allow for additional comparisons. If there are any confounds in HCP which make this difficult, that should be disclosed as these would also presumably affect the comparison with the structural and functional connectivity measures from HCP.

RESPONSE TO COMMENT 13:

In the original paper where the MIND method was initially developed, Sebenius et al executed extensive analyses to validate the technique. The primary analysis was performed using the Adolescent Brain Cognitive Development (ABCD) dataset, which includes 10,367 individuals aged 9 to 11 years old (Hagler et al. Neuroimage 2019, PMID: 31415884). Multiple additional analyses were also performed in the HCP-Young adult (HCP-YA) cohort including 960 individuals aged 21 to 35 years (van Essen et al. Neuroimage 2013, PMID: 23684880), and also in the HCP Project Development cohort, including 655 participants aged 8-21 years (Somerville et al. Neuroimage 2018, PMID: 30142446).

In Isaac Sebenius' paper, the correlations between MIND nodal-strength in the Desikan-Killiany 68-cortical regions, Desikan-Killiany 318-cortical regions and HCP parcellations for the MIND method were between 0.5 and 0.7 both in the ABCD and in the HCP-YA datasets. Similarly, edge consistency across parcellations was between 0.6 and 0.8 both in the ABCD and HCP-YA cohorts. In addition, the edge-wise correlation in MIND between the ABCD and the HCP-YA cohorts was 0.9. These results support the validity and reproducibility of the MIND approach.

In their original paper Sebenius et al. also analyzed how MIND networks recapitulated cortical microstructure investigated through the similarity between regions of the same Von Economo cytoarchitectonic classes, again showing that in both the ABCD and HCP cohorts MIND networks had substantially higher intraclass connectivity than MSNs.

The sensitivity of MIND networks to detect developmental variation was investigated as well in the original MIND paper, looking at age prediction in the HCP-YA and HCP-D cohorts, and benchmarking these methods against diffusion tractography. The authors found that MIND performed better than diffusion tensor imaging or MSNs both at the nodal strength and connection strength in the HCP-YA and HCP-D cohorts.

In summary, extensive validation of the performance of MIND networks was developed in the paper where the technique was first described. MIND showed substantially better results than MSNs or diffusion in different cohorts and through different parcellation resolutions across multiple experiments. In our paper we did not intend to perform an additional technical validation of MIND but rather we wanted to apply it across HD populations.

However, we agree that these important details were omitted in the first version of the manuscript. Therefore here we have expanded on the MIND method and its validation in the original paper by Sebenius et al.

We are copying here the text from the manuscript again to facilitate the review process.

“The Morphometric INverse Divergence (MIND) technique is a new method to investigate global brain connectivity that also leverages multiple imaging parameters at the vertex-wise level. In MIND, the similarity between brain regions is estimated through the Kullback-Leibler divergence between their distributions. The MIND method has been successfully applied to different cohorts being more consistent with principles of cortical organization than other methods such as morphometric similarity networks (MSNs). In healthy brains, highly connected regions have similar transcriptional profile. A gene co-expression network derived from the Allen Human Brain Atlas (AHBA) showed a stronger correlation with MIND-derived connectomes compared to diffusion tensor imaging or MSNs. In addition, MIND-derived connectivity metrics were more closely associated with twin-based and SNP-based heritabilities than other imaging phenotypes, particularly in primary sensory regions. Brain networks estimated through MIND are correlated with axonal connectivity and cortical cytoarchitecture, being also more sensitive to predict age than similar techniques through machine learning models. Importantly, MIND can be used with imaging features obtained from simple T1-weighted MRI sequences such as cortical thickness, surface area, mean curvature, sulcal depth and gray matter volume. These characteristics support the use of MIND to investigate connectivity across the time course of in HD.” (From page 7 line 151 to page 8 line 167)

Importantly, as previously mentioned we have adjusted all analyses for age, gender and site, to prevent these covariates having a substantial impact in the results. Moreover, ComBat harmonization did not reveal considerable effects from the site.

COMMENT 14:

The authors disclose in their discussion that they did not use any functional or diffusion imaging from the HD cohort because of limited sample sizes and heterogenous scanning protocols. I take this to mean that the three studies included here did not obtain any functional or diffusion imaging. While the argument for not including heterogenous studies makes sense, more details should be provided to bolster this argument in order to justify why COMBAT and other harmonization tools would not work for getting around these issues. This work would certainly be bolstered by including additional functional and diffusion analysis in HD, and it seems the authors do not pay much mind to this rather significant exclusion of modalities (especially given the use of functional and diffusion derivatives from normative studies!).

RESPONSE TO COMMENT 14:

Many thanks for this. Our objective was to determine the change in connectivity over time, in HD cohorts from more than 20 years before clinical motor onset until the presence of functional decline. This would not be possible if functional connectivity

and structural connectivity were included in the analysis, due to its limited availability and different acquisition parameters in the different cohorts.

In TrackHD diffusion data was only available for the last time point and not in all study sites. No fMRI data was collected in TrackHD. For TrackOn while fMRI and diffusion MRI were available, the diffusion sequence differed from that used at the last timepoint in Track-HD. Diffusion sequences also differed between TrackOn and HD-YAS, were TrackOn collected mostly single shell data, while HD-YAS collected a multishell sequence.

In addition, our group has extensively investigated diffusion and functional imaging in HD and their relationship to normative cortical datasets, such as the Allen Human Brain Atlas. For instance, we studied the relationship between gene expression and diffusion MRI in HD, revealing that white matter connectivity loss is associated with the expression of synaptic and metabolic genes (McColgan et al. *Biological Psychiatry* 2018, PMID: 29174593). Similarly, we demonstrated that cortical diffusivity in preHD is correlated with developmental as well as with synaptic and metabolic genes (Estevez-Fraga et al. *Brain* 2023, PMID: 37587097). In the HD-YAS cohort we also revealed that the relationship between fMRI connectivity and Neurofilament Light (NfL) is modulated by the spatial pattern of neuronal gene expression in the healthy brain (McColgan et al. *Brain* 2022 PMID: 35758263).

We have now try to delineate this limitation more clearly in the discussion:

“Potentially, DWI-metrics or fMRI indexes could be included. However, not all participants in our study had diffusion or functional imaging data, which would limit the interpretability of our findings. Moreover, the differences in acquisition parameters might have a potential impact in these results. Other large datasets with high quality T1-images following standardized acquisition techniques are available, but only smaller cohorts usually with heterogeneous acquisition parameters have other imaging sequences.”(Page 33, lines 677 to 684)

In addition, the ComBat harmonization method has been applied to the study populations without significant differences in MIND nodal strength in any of the 68 examined Regions-of-interest (ROIs) before or after processing. Since ComBat processing of the data did not result in a significant change we decided to use the raw data across the different analyses. We included a Figure with the ComBat pre-post results in the supplementary (Figure S1).

We have also tried to make this more clear in the results section:

“ComBat harmonization, an empirical Bayesian method equivalent to batch effect correction which removes systematic differences that are caused by different sites or scanners did not reveal significant differences in any brain region (Figure S1, Table SX). Therefore raw data was used in this study. Site was added as a covariate in all analyses.” (Page 11 lines 226 to 230)

REVIEWER 2

REMARKS TO THE AUTHOR

COMMENT TO REMARKS TO THE AUTHOR

Many thanks for this.

REMARKS ON CODE AVAILABILITY

There is a README file available; however, it is quite sparse. All of the python code is provided in notebooks with hard-coded paths to data, and no information is provided about filepath organization, CSV file format, or any other requirements that are needed to run this code. Furthermore, no documentation is provided in the python notebooks explaining what each block of code does. The R and MATLAB scripts have some better documentation; however, also hard-code filepaths without any documentation describing how data should be stored in those paths. Inline documentation is provided in the R and MATLAB files, which helps some.

Without example data or much of an idea how data was to be organized, I was not able to run the code or reproduce any results. While data release might not be possible, I would encourage the authors to include an example data set from an open-source location such as OpenNeuro.org, and to provide instructions with how to run each part of the code individually. Better yet, the authors could provide some information about how to produce specific analyses performed in their work.

COMMENT ON REMARKS ON CODE AVAILABILITY:

Many thanks for these helpful comments.

As mentioned by Reviewer #2, we could not share the original data from TrackHD, TrackOn-HD and HD-YAS, this would require a formal request by a qualified investigator to the principal investigators and funders. Therefore, we have also provided CSV files with random numbers reflecting the original biological variables (e.g. age between 40 and 60 years). This dummy CSV files can be used to run the scripts provided.

In this new version we also have provided a CSV examples of MIND matrices (XXX/XXX) with files from different subjects, anonymizing their IDs and rest of the data. *Cohen's d* for nodal strength can be generated using these files together with the scripts provided. We also provided the final results for the Cohens's *d* across regions in all cohorts, that can be used to run additional scripts.

Now we have rewritten all the scripts provided to: 1) Improve readability 2) Reference the original sources 3) Make it possible, and easier to run the code

In the new GitHub repository we have also included scripts showing how the brain surface images were generated, as well as how we performed ComBat harmonization and additional parcellations for robustness.

All these steps have been explained in detail in the new README file with a detailed explanation of the input data required and output obtained in each step.

REVIEWER 3

REMARKS TO THE AUTHOR:

In this manuscript, the authors used Morphometric INverse Divergence (MIND) to study the degree of divergence in connectivity patterns in controls versus Huntington's Disease (HD) patients along the disease progression time course including early premanifest, late premanifest, and manifest HD. Utilizing T1-weighted MRI data from three HD study cohorts HD-YAS (57 early preHD, 60 controls), TrackOn-HD (85 late preHD, 89 controls), and Track-HD (110 manifest HD and 111 controls), the authors generated MIND networks. Group-level analysis results of these data revealed disease epicenters and correlations with plasma NFL measurements, and neurotransmitter distributions were performed to study organizational principles, receptor gradients, and neurotransmitter systems.

This is an interesting paper suitable for publication in Nature Communications. The data presented in this paper support the main conclusion and reveal interesting details of connectivity alterations in HD.

COMMENT TO REMARKS TO THE AUTHOR:

Many thanks for these kind comments.

COMMENT 1

The manuscript would be improved by considering the following points:

Page 6

Lines 11 -16

"The Morphometric INverse Divergence (MIND) technique is a new method to investigate brain connectivity that leverages multiple imaging features obtained from T1-weighted MRI such as cortical thickness, surface area, mean curvature, sulcal depth and gray matter volume. MIND-derived brain networks are closely associated with biological characteristics such as gene expression, being a useful method to investigate the pathophysiological mechanisms involved in connectivity alterations in HD17."

It is unclear from this paragraph, yet it is an important point for this paper, why MIND-derived networks are closely associated with gene expression. The reader would benefit from more information and details on the original MIND reference #17, in which the term was coined.

RESPONSE TO COMMENT 1:

In this new version of the manuscript we have tried to expand on the development of the MIND method. Specifically we have added more details about the association between MIND connectivity and gene expression

"The Morphometric INverse Divergence (MIND) technique is a new method to investigate global brain connectivity that also leverages multiple imaging parameters at the vertex-wise level. In MIND, the similarity between brain regions is estimated through the Kullback-Leibler divergence between their distributions. The MIND method has been successfully applied to different cohorts being more consistent

with principles of cortical organization than other methods such as morphometric similarity networks (MSNs). In healthy brains, highly connected regions have similar transcriptional profile. A gene co-expression network derived from the Allen Human Brain Atlas (AHBA) showed a stronger correlation with MIND-derived connectomes compared to diffusion tensor imaging or MSNs. In addition, MIND-derived connectivity metrics were more closely associated with twin-based and SNP-based heritabilities than other imaging phenotypes, particularly in primary sensory regions. Brain networks estimated through MIND are correlated with axonal connectivity and cortical cytoarchitecture, being also more sensitive to predict age than similar techniques through machine learning models. Importantly, MIND can be used with imaging features obtained from simple T1-weighted MRI sequences such as cortical thickness, surface area, mean curvature, sulcal depth and gray matter volume. These characteristics support the use of MIND to investigate connectivity across the time course of in HD.” (From page 7 line 151 to page 8 line 167)

COMMENT 2

Lines 20-21

“We hypothesized the existence of hyperconnectivity in HD participants very far from disease onset, progressing to hypoconnectivity through later stages.”

Specify “very far” or refrain from making an emphasized yet vague statement.

RESPONSE TO COMMENT 2

We used the term ‘far from disease onset’ based on the terminology used in the original HD-YAS paper (Scahill et al. Lancet Neurology 2020 PMID: 32470422). However, we agree that it subjective and we have therefore changed it to the number of years to onset:

“We hypothesized the existence of hyperconnectivity in HD participants 22 years from disease onset” (Page 8 lines 170-171)

“We demonstrate a connectivity gradient across the disease time course of HD, with hyperconnectivity occurring 22 years from disease onset” (Page 8 lines 185-186)

COMMENT 3

Page 9

Lines 13-14

“Robustness analyses were performed to examine the impact of ComBat harmonization as well as different parcellation resolutions (Figures S1-S2)”
Consider explaining ComBat harmonization as equivalent to batch effect correction, removing systematic differences that are caused by different sites or scanners for example, while leaving biological signal intact.

RESPONSE TO COMMENT 3:

We have now expanded on the definition of ComBat harmonization in the Results section, also adding that Site has been included as a covariate in all analyses. We also added reference to the original paper where ComBat was developed.

“Combat harmonization, an empirical Bayesian method equivalent to batch effect correction which removes systematic differences that are caused by different sites or

scanners did not reveal significant differences in any brain region (Figure S1). Therefore raw data was used throughout. Site was added as a covariate in all analyses.” (Page 11, lines 226-230)

COMMENT 4:

Page 10
Lines 1-3

“For connection strength (Figure 2, Tables S2-S5) in early preHD, 21.67 years from disease onset, NBS analyses revealed significant increases in connectivity relative to healthy controls (PFWE=0.029); no decreases in connectivity were observed.”
The authors list all significant and non-significant p-values throughout their manuscript. Consider adding this non-significant p-value to the text as well.

RESPONSE TO COMMENT 4:

In line with Comment 9 from Reviewer #1 we have removed all non-significant P values from the body of the paper referring to them as ‘non significant’ in the text. However, we have now added all P-values in the following tables in the supplementary:

- A table with the raw P/FDR-corrected values before / after ComBat harmonization (Table S2)
- A table with the raw/FDR-corrected P_{spin} values from the epicentre analysis. (Table S22)
- A table with the raw/FDR-corrected P_{spin} values from the organizational principles analysis (Table S23)
- A table with the raw/FDR-corrected P_{spin} values from the PET neurotransmitter analysis (Table S25)
- A table with the raw/FDR-corrected P_{spin} values from the autoradiography neurotransmitter analysis (Table S26)

COMMENT 5

Lines 6-7

“Increased connectivity was also seen in five connections in mHD (PFWE=0.022), localized to the occipital cortex (Figure S3).”
Figure S3 shows the hyperconnectivity of four occipital cortical regions and one parietal (superior parietal) region. Please correct the statement accordingly.

RESPONSE TO COMMENT 5:

Thanks for noticing this. There are four connections within the occipital cortex and one between the superior parietal and the pericalcarine cortex. We have now amended the text to:

“Increased connectivity was also seen in five connections in mHD ($P_{\text{FWE}}=0.022$), four occipito-occipital and one occipito-parietal connection (Figure S3)” (Page 13 Lines 271-273)

COMMENT 6

“These findings strongly suggest that the areas with increased connectivity during very early stages of the disease are the same brain regions that then experience connectivity loss later in the disease course.”

If these findings strongly suggest that the same areas that show increased connectivity in early preHD are the same that experience connectivity loss later in the disease - then couldn't / shouldn't there be an observable and significant inverse correlation between node strength in early preHD and mHD?

RESPONSE TO COMMENT 6:

Thanks for this. We found indeed correlations between early preHD and late preHD and also between late preHD and mHD. However, the correlation between early preHD and mHD was not significant. We have therefore toned down the wording of the sentence mentioned by Reviewer #3 removing the adjective “strongly” . We have also adapted the made it clear that the areas experiencing hyperconnectivity early on are not exactly “the same” but “similar” to the ones developing connectivity decreases in later stages:

“These findings suggest that the areas with increased connectivity during very early stages of the disease are the same brain regions that then experience connectivity loss later in the disease course.” (Page14 Lines 296-298)

The same change has been implemented in the discussion

“As the disease advances, hyperconnectivity progresses to hypoconnectivity affecting similar areas” (Page 29 Lines 537-538)

In addition, with the aim of describing in more detail the nature of the associations between nodal strength and connection strength between cohorts we have now added the tables below to the supplementary. Here, we outline the correlations coefficients and P values between the different pairs of cohorts(Table SXX).

Table S12
NODAL STRENGTH

Cohorts	Rho	P value	P_{FDR}
mHD – late preHD	0.382	0.001	0.04605
mHD – early preHD	0.007	0.952	0.9520
Late preHD-early preHD	0.262	0.0307	0.00300

Table S13
CONNECTION STRENGTH

Cohorts	Rho	P value	P _{FDR}
mHD – late preHD	0.386	< 0.00001	< 0.00001
mHD – early preHD	0.119	< 0.00001	< 0.00001
Late preHD-early preHD	0.235	< 0.00001	< 0.00001

We have also developed a new figure (Figure S4) with scatterplots depicting the associations between nodal strength or connection strength and the different cohorts (e.g. illustrating the association between nodal strength in early preHD and nodal strength in mHD).

COMMENT 7:

Page 14

Lines 4-6 and lines 27-29

“In addition, the structural and functional connectivity patterns of specific regions (“epicentres”) has been proposed to align with disease-associated atrophy, indicating that transneuronal spread of pathogenic factors contributes to neural death in neurodegenerative diseases such as frontotemporal dementia or Alzheimer’s disease 28.”

“These results suggest that the relative contribution of axonal connections varies during the course of the disease, being particularly prominent during late preHD stages, when potentially pathogenic factors such as the misfolded mHTT could spread along axons in highly connected regions²⁹. This phenomenon is region-specific, targeting paracentral and cingulate cortices, which are known to develop early cortical gray matter loss in HD²³.”

Also page 17

Lines 18-20

“Shortly before motor onset, axonal spread of pathogenic factors could contribute to accelerated atrophy.”

As there is no direct and definite proof for the spread of pathogenic factors along axons, alternative hypotheses should also be discussed. For example, synaptic dysfunction, leading to aberrant neuronal signalling, can result in degeneration.

RESPONSE TO COMMENT 7:

We agree with the reviewer and have mentioned aberrant synaptic signalling and/or other factors as potential contributors to contiguous degeneration along highly connected regions.

“The presence of these hubs suggest that either transneuronal spread of pathogenic factors or other mediators such as synaptic dysfunction, contribute to neural death in neurodegenerative diseases such as frontotemporal dementia or Alzheimer’s disease” (Page 17, lines 326-329)

“These results suggest that the relative contribution of axonal connections varies during the course of the disease, being particularly prominent during late preHD stages, when potentially pathogenic factors such as the misfolded mHTT could spread along axons in highly connected region. However, additional mechanisms such as synaptic dysfunction leading to aberrant signalling and eventual degeneration might also be involved, either independently or alongside transmission of pathogenic factors along axons. These phenomena are region-specific, targeting paracentral and cingulate cortices, which are known to develop early cortical gray matter loss in HD.” (Page 18, lines 353-360)

“Shortly before motor onset, axonal spread of pathogenic factors or other biological events such as abnormal synaptic signalling contribute to accelerated atrophy” (Page 21, lines 417-419)

COMMENT 8

Page 19

Lines 6-8

“PET neurotransmitter data for 19 receptors and transporters from more than 1,200 healthy controls was parcellated using the Desikan-Killiany atlas²⁶ obtaining a 68-brain region by 19-neurotransmitter matrix¹⁵.”

Please list here the neurotransmitter families comprising these 19 neurotransmitters (Serotonin, Dopamine, GABA, etc.)

RESPONSE TO COMMENT 8:

We have now included the neurotransmitter families as well as the different receptors and transporters as suggested by Reviewer #3:

“PET neurotransmitter data for 19 receptors and transporters from more than 1,200 healthy controls was parcellated using the Desikan-Killiany atlas obtaining a 68-brain region by 19-neurotransmitter matrix and compared with HD-specific nodal strength. Included neurotransmitters and transporters were Serotonin (5-HT_{1A}, 5-HT_{1B}, 5-HT_{2A}, 5-HT₄, 5-HT₆, 5-HTT), dopamine (D₁, D₂, DAT), norepinephrine (NET), histamine (H₃) Acetylcholine ($\alpha_4\beta_2$, M₁, VACHT), cannabinoid (CB₁), opioid (MOR), glutamate (NMDA, mGluR₅) and GABA (GABA_{A/BZ}).” (Page 22, lines 436-442)

COMMENT 9

Page 25

Lines 20-23

“Our previous work has shown that changes in the occipital cortex of pwHD are largely developmental rather than neurodegenerative³³ and occipital hyperconnectivity could be related to different mechanisms governing regions densely connected to the striatum, or compensation.”

Consider rewording for clarity. Suggestion:

“Our previous work has shown that changes in the occipital cortex of pwHD are largely developmental rather than neurodegenerative³³ and occipital hyperconnectivity could be related to compensation, or different mechanisms governing regions densely connected to the striatum.”

RESPONSE TO COMMENT 9:

Suggestion accepted and implemented in the manuscript:

“Our previous work has shown that changes in the occipital cortex of pwHD are largely developmental rather than neurodegenerative and occipital hyperconnectivity could be related to compensation, or different mechanisms governing regions densely connected to the striatum.” (Page 30, Lines 581-584)

COMMENT 10:

Page 26

Lines 25-27

“Interestingly, a recent study has shown that medium spiny neurons, but not other striatal cells, undergo somatic expansion, which subsequently triggers severe changes in gene expression in these neurons leading to neuronal death and atrophy³⁹.”

A second study, already mentioned in the manuscript as reference number 51 (Mätlik K, Baffuto M, Kus L, et al. Cell-type-specific CAG repeat expansions and toxicity of mutant Huntingtin in human striatum and cerebellum. *Nat Genet* 2024; 56: 383– 94.) , should be added to this statement alongside reference number 39 (Handsaker RE, Kashin S, Reed NM, et al. Long somatic DNA-repeat expansion drives neurodegeneration in Huntington disease. *bioRxiv* 2024; : 2024.05.17.592722.).

RESPONSE TO COMMENT 10:

We agree with the reviewer and have added the suggested citation to the sentence.

COMMENT 11

Figure 1

Please organize sections in marked figure panels (A-Z) and match with text for improved interpretability and readability.

RESPONSE TO COMMENT 11:

We have added figure panels to Figure 1, and referenced them in the figure legend. We have also referenced the specific panel in Figure 1 throughout the text, wherever each analysis is performed.

“Figure 1 Summary of analysis pipeline and main results

Data from three cohorts (a) were used in this study: HD-YAS (57 early preHD, 60 controls), TrackOn-HD (85 late preHD, 89 controls) and Track-HD (110 manifest HD and 111 controls). Structural T1-weighted brain MRI scans (b) were parcellated using the 68 cortical region Desikan-Killiany atlas with Freesurfer’s version 7 recon-all command (c). Cortical thickness, mean curvature, surface area, sulcal depth and gray matter volume were obtained (d) to compute subject-specific MIND networks.

A linear mixed model was applied to each cohort obtaining z-scored effect sizes (Cohen’s d) in patient populations versus healthy controls for nodal strength and connection strengths across MIND cortical similarity matrices(e). There was hyperconnectivity in HD participants two decades before motor onset, progressing to hypoconnectivity during later stages (f), the latter being associated with neuroaxonal damage (g).

Healthy structural and functional connectomes were spatially correlated with nodal strength to investigate the presence of disease epicentres (h). The distribution of neurotransmitter receptors and transporters was computed using PET data from more than 1,200 healthy individuals using data from Hansen et al. (2022) (i). The relative contribution of additional structural, functional and cell-autonomous organizational principles was explored (j), finding that cell-autonomous gene expression and neurotransmitter distribution are major mechanisms contributing to nodal strength. Receptome analysis (k) showed that the first component of the distribution of neurotransmitter systems was associated with nodal strength. Finally, a neurotransmitter receptor analysis was performed disclosing a predominant association between the distribution of cholinergic and serotonergic systems with nodal strength (l). The association with neurotransmitter systems was more prominent in granular and infragranular cortical layers.

HD, Huntington’s disease; MIND, Morphometric Inverse Divergence; preHD, premanifest HD.” (Page 10 lines 198-209)

“512 participants were included, comprising 252 pwHD and 260 controls. These data was obtained from three different studies: HD-YAS (57 early preHD and 60 controls), TrackOn-HD (85 late preHD and 89 controls) and Track HD (110 mHD and 111 controls) (Figure 1A).” (Page 11, lines 212-217)

“T1-weighted MRI-derived structural features parcellated through Freesurfer’s version 7 recon-all command were processed using the MIND approach (Figure 1 B-E)” (Page 12, lines 240-241)

“estimated through adjusted Cohen’s d, providing a numeric value for the node strength and connection strength difference between pwHD and controls from each cohort (HD-YAS -early preHD-, TrackOn-HD -late preHD- and Track-HD -mHD-) in each node and in each connection.” (Page 12, lines 254-256)

“Group-wise differences between pwHD and controls were explored for both connection strength and node strength, across cohorts. For connection strength and correlations with NfL concentrations (Figure 1G),” (Page 12, lines 259-265)

“To investigate the presence of disease epicentres in HD (Figure 1H),” (Page 17, line 10)

“We explored associations with different organizational principles, obtained from datasets of healthy controls different from the HD-YAS, TrackOn-HD and Track-HD cohorts (Figure 1J).” (Page 20, lines 329-332)

“PET neurotransmitter data for 19 receptors and transporters from more than 1,200 healthy controls was parcellated using the Desikan-Killiany atlas obtaining a 68-brain region by 19-neurotransmitter matrix and compared with HD-specific nodal strength (Figure 1K).” (Page 23, lines 436-439)

“PET maps from healthy controls were investigated and association with HD-specific nodal strength was computed through the $R^2_{adjusted}$ of each multilinear regression model was tested using FDR-corrected spin tests (Figure 1I, 1L).” (Page 26, lines 482-484)

COMMENT 12:

Figure 2

Resolution low, small font hardly readable, i.e. in panel a (group network connection strength values) and e (cortical region annotations).

RESPONSE TO COMMENT 12 :

Apologies for this. In the initial version of the manuscript we embedded the PNG figures in the manuscript with a resolution of 300 dpi. However, when these are embedded into Word the resolution decreases, which makes we agree makes it difficult to read some panels, particularly circular graphs.

In this new version we have uploaded all figures separately in PDF format with high resolution. It should now be possible to zoom in as much as needed and read any text present in the figures. We have also updated with Figure 3 high resolution panels

COMMENT 13:

Redundant word “were” in the figure text:

d, Violin plots depicting ROIs showing significant differences at the uncorrected level in early preHD and late preHD. In mHD, for visualization purposes, only the top five ROIs with larger effect sizes were shown, all of them significant following FDR correction at a $P < 0.05$.

RESPONSE TO COMMENT 13:

Thanks for noticing this. We have removed the word “were”:

“Violin plots depicting ROIs showing significant differences at the uncorrected level in early preHD and late preHD. In mHD, for visualization purposes, only the top five ROIs with larger effect sizes were shown,” (Page 15 lines 307-308)

COMMENT 14:

Figure 3

“b, Scatterplots representing the correlations between plasma NfL and nodal strength for MIND connectivity in early preHD, late preHD and mHD. The ROI with largest absolute magnitude in the correlation coefficient was selected for each cohort for visualisation purposes.”

The selection of the ROIs with the largest absolute magnitude for visualization purposes is unclear. Appreciating the presentation of the distribution of the individual data points that form the largest correlation coefficients and underlie the brain heat map extremes in panel a, I suggest elaborating on and explaining better the intention of the “visualization purposes” in the figure text.

RESPONSE TO COMMENT 14:

We have reflected on this and think that representing a single ROI out of a total of 68 is not really informative, particularly when the correlation coefficients are presented in panel A of the same figure, and the significant connections in panels C-D (now B-C)

We have therefore removed the Figure 3 scatterplots from this new version and modified the figure legend accordingly.

COMMENT 15:

General

Please add to the discussion a comment on how a left-right distribution of significant associations between connectivity and plasma NfL as shown in Figure 3, panel c can be explained.

RESPONSE TO COMMENT 15

There is extensive evidence showing that inter-hemispheric connections are affected early during the course of HD (McColgan et al JCI Insight, 2017 PMID: 28422761; McColgan et al Biological Psychiatry, 2017 PMID: 29174593). This may be related to the loss of cortical intratelencephalic neurons that project to the striatum and across the corpus callosum (McColgan Nature Reviews Neuroscience 2020).

In keeping with this, the corpus callosum, the main conduit of information between cerebral hemispheres has shown early involvement with a temporally and topologically selective pattern of degeneration in HD both using structural (Crawford et al. J Huntington's disease, 2013, PMID: 25062736) and diffusion MRI (Diana Rosas et al. Neuroimage, 2010 PMID: 19850138).

We have now added this information to the discussion:

“There is extensive evidence showing that inter-hemispheric connections are affected early during the course of HD. This may be related to the loss of cortical intratelencephalic neurons that project to the striatum and across the corpus callosum In keeping with this the corpus callosum, the main conduit of information between cerebral hemispheres has shown early involvement with a temporally and topologically selective pattern of degeneration in HD both using structural and diffusion MRI.” (Page 31 lines 592-597)

COMMENT 16:

Figure 4

General

Given the pronounced vulnerability and degeneration observed in the striatum in Huntington's Disease, the absence of distinct subcortical epicenters is unexpected. This observation raises important questions about underlying disease mechanisms or potential methodological factors, warranting further discussion by the authors.

RESPONSE TO COMMENT 16:

The absence of subcortical epicentres suggests that degeneration of cortico-cortical connections may be either independent or temporally dissociated from subcortical pathology. Indeed the very early occurrence of striatal pathology, seen 24 years before disease onset in HD suggests the latter. The absence of subcortical epicentres may also be related to methodological factors such as difficulty in the accurate segmentation of small subcortical nuclei using MRI methods.

We have added a paragraph outlining all these factors in the discussion, also mentioning that our results could potentially be related to methodological limitations:

“Interestingly, there was no association between subcortical epicentres in healthy controls and cortical MIND networks in HD. This suggests that degeneration of cortico-cortical connections may be either independent or temporally dissociated from subcortical pathology. Indeed the very early occurrence of striatal pathology, seen 24 years before disease onset in HD (include YAS ref) suggests the latter. The absence of subcortical epicentres may also be related to methodological factors such as difficulty in the accurate segmentation of small subcortical nuclei using MRI methods.” (Page 31 lines 612-618)

REVIEWER 4

Reviewer #4 (Remarks to the Author):

RESPONSE T REMARKS TO THE AUTHOR:

Many thanks for this.

REVIEWERS' COMMENTS

REVIEWER 1

REMARKS TO THE AUTHOR

The revised manuscript is excellent. The authors responded to our previous review's input and have strengthened the paper. These findings are presented more clearly and sufficiently cited. Any remaining limitations in this research will be addressed by what will hopefully be a strategic response from the field to embrace this paper and follow on to advance understanding of HD further using connectivity and neuroimaging. Congratulations.

RESPONSE TO REMARKS TO THE AUTHOR

Many thanks for this. We agree that the paper has improved significantly following the suggestions from the reviewers.

REMARKS ON CODE AVAILABILITY

Not applicable.

RESPONSE TO REMARKS ON CODE AVAILABILITY

Not applicable.

REVIEWER 2

REMARKS TO THE AUTHOR

RESPONSE TO REMARKS TO THE AUTHOR

Not applicable.

REVIEWER 3

REMARKS TO THE AUTHOR

I have carefully reviewed the revisions made by the authors in response to the comments provided during the initial review.

The authors have satisfactorily addressed the concerns raised and made significant improvements to the manuscript.

I believe this revised version meets the standards for publication and represents an important contribution to the field.

However, I recommend correcting the formatting of Table S22 P values in the epicenter analysis, before the manuscript is published.

RESPONSE TO REMARKS TO THE AUTHOR

Many thanks for your comments. We have now amended Table S22

REVIEWER 4

REMARKS TO THE AUTHOR

RESPONSE TO REMARKS TO THE AUTHOR

Not applicable.